Manuscript prepared for Atmos. Chem. Phys.
with version 2015/04/24 7.83 Copernicus papers of the LATEX class copernicus.cls.
Date: 30 May 2016

# A multi-model intercomparison of halogenated very short-lived substances (TransCom-VSLS): linking oceanic emissions and tropospheric transport for a reconciled estimate of the stratospheric source gas injection of bromine

R. Hossaini[1,a], P. K. Patra[2], A. A Leeson[1,b], G. Krysztofiak[3,c], N. L. Abraham[4,5], S. J. Andrews[6], A. T. Archibald[4], J. Aschmann[7], E. L. Atlas[8], D. A. Belikov[9,10,11], H. Bönisch[12], R. Butler[13], L. J. Carpenter[6], S. Dhomse[1], M. Dorf[14], A. Engel[12], L. Feng[13], W. Feng[1,4], S. Fuhlbrügge[15], P. T. Griffiths[5], N. R. P. Harris[5], R. Hommel[7], T. Keber[12], K. Krüger[15,16], S. T. Lennartz[15], S. Maksyutov[9], H. Mantle[1], G. P. Mills[17], B. Miller[18], S. A. Montzka[18], F. Moore[18], M. A. Navarro[8], D. E. Oram[17], P. I. Palmer[13], K. Pfeilsticker[19], J. A. Pyle[4,5], B. Quack[15], A. D. Robinson[5], E. Saikawa[20,21], A. Saiz-Lopez[22], S. Sala[12], B.-M Sinnhuber[3], S. Taguchi[23], S. Tegtmeier[15], R. T. Lidster[6], C. Wilson[1,24], and F. Ziska[15]

[1]School of Earth and Environment, University of Leeds, Leeds, UK.
[2]Department of Environmental Geochemical Cycle Research, JAMSTEC, Yokohama, Japan.
[3]Institute for Meteorology and Climate Research, Karlsruhe Institute of Technology, Karlsruhe, Germany.
[4]National Centre for Atmospheric Science, UK.
[5]Department of Chemistry, University of Cambridge, Cambridge, UK.
[6]Department of Chemistry, University of York, Heslington, York, UK.
[7]Institute of Environmental Physics, University of Bremen, Bremen, Germany.
[8]Rosenstiel School of Marine and Atmospheric Science, University of Miami, USA.
[9]Center for Global Environmental Research, National Institute for Environmental Studies, Tsukuba, Japan.
[10]National Institute of Polar Research, Tokyo, Japan.
[11]Tomsk State University, Tomsk, Russia.
[12]Institute for Atmospheric and Environmental Sciences, Universität Frankfurt/Main, Germany.
[13]School of GeoSciences, The University of Edinburgh, Edinburgh, UK.
[14]Max-Planck-Institute for Chemistry, Mainz, Germany.
[15]GEOMAR Helmholtz Centre for Ocean Research Kiel, Kiel, Germany.
[16]University of Oslo, Department of Geosciences, Oslo, Norway.
[17]School of Environmental Sciences, University of East Anglia, Norwich, UK.
[18]National Oceanic and Atmospheric Administration, Boulder, USA.
[19]Institute for Environmental Physics, University of Heidelberg, Heidelberg, Germany.
[20]Department of Environmental Sciences, Emory University, Atlanta, USA.
[21]Department of Environmental Health, Rollins School of Public Health, Emory University, Atlanta, USA.
[22]Atmospheric Chemistry and Climate Group, Institute of Physical Chemistry Rocasolano, CSIC, Madrid, Spain.
[23]National Institute of Advanced Industrial Science and Technology, Japan.
[24]National Centre for Earth Observation, Leeds, UK.

[a]now at: Department of Chemistry, University of Cambridge, Cambridge, UK.
[b]now at: Lancaster Environment Centre, University of Lancaster, Lancaster, UK.
[c]now at: Laboratoire de Physique et Chimie de l'Environnement et de l'Espace, CNRS-Université d'Orléans, Orléans, France.

*Correspondence to:* Ryan Hossaini
(r.hossaini@leeds.ac.uk)

**Abstract.** The first concerted multi-model intercomparison of halogenated very short-lived substances (VSLS) has been performed, within the framework of the ongoing Atmospheric Tracer Transport Model Intercomparison Project (TransCom). Eleven global models or model variants participated (nine chemical transport models and two chemistry-climate models) by simulating the major natural bromine VSLS, bromoform ($CHBr_3$) and dibromomethane ($CH_2Br_2$), over a 20-year period (1993-2012). Except for 3 model simulations, all others were driven offline by (or nudged to) re-analysed meteorology. The overarching goal of TransCom-VSLS was to provide a reconciled model estimate of the stratospheric source gas injection (SGI) of bromine from these gases, to constrain the current measurement-derived range, and to investigate inter-model differences due to emissions and transport processes. Models ran with standardised idealised chemistry, to isolate differences due to transport, and we investigated the sensitivity of results to a range of VSLS emission inventories. Models were tested in their ability to reproduce the observed seasonal and spatial distribution of VSLS at the surface, using measurements from NOAA's long-term global monitoring network, and in the tropical troposphere, using recent aircraft measurements - including high altitude observations from the NASA Global Hawk platform.

The models generally capture the observed seasonal cycle of surface $CHBr_3$ and $CH_2Br_2$ well, with a strong model-measurement correlation ($r \geq 0.7$) at most sites. In a given model, the absolute model-measurement agreement at the surface is highly sensitive to the choice of emissions. Large inter-model differences are apparent when using the same emission inventory, highlighting the challenges faced in evaluating such inventories at the global scale. Across the ensemble, most consistency is found within the tropics where most of the models (8 out of 11) achieve best agreement to surface $CHBr_3$ observations using the lowest of the three $CHBr_3$ emission inventories tested (similarly, 8 out of 11 models for $CH_2Br_2$). In general, the models reproduce well observations of $CHBr_3$ and $CH_2Br_2$ obtained in the tropical tropopause layer (TTL) at various locations throughout the Pacific. Zonal variability in VSLS loading in the TTL is generally consistent among models, with $CHBr_3$ (and to a lesser extent $CH_2Br_2$) most elevated over the tropical West Pacific during boreal winter. The models also indicate the Asian Monsoon during boreal summer to be an important pathway for VSLS reaching the stratosphere, though the strength of this signal varies considerably among models.

We derive an ensemble climatological mean estimate of the stratospheric bromine SGI from $CHBr_3$ and $CH_2Br_2$ of 2.0 (1.2-2.5) ppt, $\sim$57% larger than the best estimate from the most recent World Meteorological Organization (WMO) Ozone Assessment Report. We find no evidence for a long-term, transport-driven trend in the stratospheric SGI of bromine over the simulation period. The transport-driven inter-annual variability in the annual mean bromine SGI is of the order of $\pm$5%, with SGI exhibiting a strong positive correlation with ENSO in the East Pacific. Overall, our results do not show systematic differences between models specific to the choice of reanalysis me-

teorology, rather clear differences are seen related to differences in the implementation of transport processes in the models.

## 1 Introduction

Halogenated very short-lived substances (VSLS) are gases with atmospheric lifetimes shorter than, or comparable to, tropospheric transport timescales ($\sim$6 months or less at the surface). Naturally-emitted VSLS, such as bromoform ($CHBr_3$), have marine sources and are produced by phytoplankton (e.g. Quack and Wallace, 2003) and various species of seaweed (e.g. Carpenter and Liss, 2000) - a number of which are farmed for commercial application (Leedham et al., 2013). Once in the at-

mosphere, VSLS (and their degradation products) may ascend to the lower stratosphere (LS), where they contribute to the inorganic bromine ($Br_y$) budget (e.g. Pfeilsticker et al., 2000; Sturges et al., 2000) and thereby enhance halogen-driven ozone ($O_3$) loss (Salawitch et al., 2005; Feng et al., 2007; Sinnhuber et al., 2009; Sinnhuber and Meul, 2015). On a per molecule basis, $O_3$ perturbations near the tropopause exert the largest radiative effect (e.g. Lacis et al., 1990; Forster and Shine, 1997;

Riese et al., 2012) and recent work has highlighted the climate relevance of VSLS-driven $O_3$ loss in this region (Hossaini et al., 2015a).

     Quantifying the contribution of VSLS to stratospheric $Br_y$ ($Br_y^{VSLS}$) has been a major objective of numerous recent observational studies (e.g. Dorf et al., 2008; Laube et al., 2008; Brinckmann et al., 2012; Sala et al., 2014; Wisher et al., 2014) and modelling efforts (e.g. Warwick et al., 2006; Hossaini et al.,

2010; Liang et al., 2010; Aschmann et al., 2011; Tegtmeier et al., 2012; Hossaini et al., 2012b, 2013; Aschmann and Sinnhuber, 2013; Fernandez et al., 2014). However, despite a wealth of research, $Br_y^{VSLS}$ remains poorly constrained, with a current best-estimate range of 2-8 ppt reported in the most recent World Meteorological Organization (WMO) Ozone Assessment Report (Carpenter and Reimann, 2014). Between 15% and 76% of this supply comes from the stratospheric *source gas injection*

(SGI) of VSLS; i.e. the transport of a source gas (e.g. $CHBr_3$) across the tropopause, followed by its breakdown and in-situ release of $Br_y^{VSLS}$ in the LS. The remainder comes from the troposphere-to-stratosphere transport of both organic and inorganic product gases, formed following the breakdown of VSLS below the tropopause; termed *product gas injection* (PGI).

     Owing to their short tropospheric lifetimes, combined with significant spatial and temporal inho-

mogeneity in their emissions (e.g. Carpenter et al., 2005; Archer et al., 2007; Orlikowska and Schulz-Bull, 2009; Ziska et al., 2013; Stemmler et al., 2015), the atmospheric abundance of VSLS can exhibit sharp tropospheric gradients. The stratospheric SGI of VSLS is expected to be most efficient in regions where strong uplift, such as convectively active regions, coincides with regions of elevated surface mixing ratios (e.g. Tegtmeier et al., 2012, 2013; Liang et al., 2014), driven by strong lo-

calised emissions or "hot spots". Both the magnitude and distribution of emissions, with respect to transport processes, could be, therefore, an important determining factor for SGI. However, current

global-scale emission inventories of $CHBr_3$ and $CH_2Br_2$ are poorly constrained, owing to a paucity of observations used to derive their surface fluxes (Ashfold et al., 2014), contributing significant uncertainty to model estimates of $Br_y^{VSLS}$ (Hossaini et al., 2013). Given the uncertainties outlined above, it is unclear how well preferential transport pathways of VSLS to the LS are represented in global scale models.

Strong convective source regions, such as the tropical West Pacific during boreal winter, are likely important for the troposphere-to-stratosphere transport of VSLS (e.g. Levine et al., 2007; Aschmann et al., 2009; Pisso et al., 2010; Hossaini et al., 2012b; Liang et al., 2014). The Asian Monsoon also represents an effective pathway for boundary layer air to be rapidly transported to the LS (e.g. Randel et al., 2010; Vogel et al., 2014; Orbe et al., 2015; Tissier and Legras , 2016), though its importance for the troposphere-to-stratosphere transport of VSLS is largely unknown, owing to a lack of observations in the region. While global models simulate broadly similar features in the spatial distribution of convection, large inter-model differences in the amount of tracers transported to the tropopause have been reported by Hoyle et al. (2011), who performed a model intercomparison of idealised ("VSLS-like") tracers with a uniform surface distribution. In order for a robust estimate of the stratospheric SGI of bromine, it is necessary to consider spatial variations in VSLS emissions, and how such variations couple with transport processes. However, a concerted model evaluation of this type has yet to be performed.

Over a series of two papers, we present results from the first VSLS multi-model intercomparison project (TransCom-VSLS). The TransCom initiative was setup in the 1990s to examine the performance of chemical transport models. Previous TransCom studies have examined non-reactive tropospheric species, such as sulphur hexafluoride ($SF_6$) (Denning et al., 1999) and carbon dioxide ($CO_2$) (Law et al., 1996, 2008). Most recently, TransCom projects have examined the influence of emissions, transport and chemical loss on atmospheric $CH_4$ (Patra et al., 2011) and $N_2O$ (Thompson et al., 2014). The overarching goal of TransCom-VSLS was to constrain estimates of $Br_y^{VSLS}$, towards closure of the stratospheric bromine budget, by (i) providing a reconciled climatological model estimate of bromine SGI, to reduce uncertainty on the measurement-derived range (0.7-3.4 ppt Br) - currently uncertain by a factor of $\sim$5 (Carpenter and Reimann, 2014) - and (ii) quantify the influence of emissions and transport processes on inter-model differences in SGI. In this regard, we define *transport* differences between models as the effects of boundary layer mixing, convection and advection, and the implementation of these processes. The project was not designed to separate clearly the contributions of each transport component in the large model ensemble, but can be inferred as the boundary layer mixing affects tracer concentrations mainly near the surface, convection controls tracer transport to the upper troposphere and advection mainly distributes tracers horizontally (e.g. Patra et al., 2009). Specific objectives were to (a) evaluate models against measurements from the surface to the tropical tropopause layer (TTL) and (b) examine zonal and seasonal variations in VSLS loading in the TTL. We also show inter-annual variability in the strato-

spheric loading of VSLS (limited to transport) and briefly discuss possible trends related to the El
Niño Southern Oscillation (ENSO). Section 2 gives a description of the experimental design and
an overview of participating models. Model-measurement comparisons are given in Sections 3.1
to 3.3. Section 3.4 examines zonal/seasonal variations in the troposphere-stratosphere transport of
VSLS and Section 3.5 provides our reconciled estimate of bromine SGI and discusses inter-annual
variability.

## 2  Methods, Models and Observations

Eleven models, or their variants, took part in TransCom-VSLS. Each model simulated the major
bromine VSLS, bromoform ($CH Br_3$) and dibromomethane ($CH_2Br_2$), which together account for
77-86% of the total bromine SGI from VSLS reaching the stratosphere (Carpenter and Reimann,
2014). Participating models also simulated the major iodine VSLS, methyl iodide ($CH_3I$), though
results from the iodine simulations will feature in a forthcoming, stand-alone paper (Hossaini et al.
2016, in prep). Each model ran with multiple $CH Br_3$ and $CH_2Br_2$ emission inventories (see Section
2.1) in order to (i) investigate the performance of each inventory, in a given model, against observa-
tions and (ii) identify potential inter-model differences whilst using the same inventory. Analogous
to previous TransCom experiments (e.g. Patra et al., 2011), a standardised treatment of tropospheric
chemistry was employed, through use of prescribed oxidants and photolysis rates (see Section 2.2).
This approach (i) ensured a consistent chemical sink of VSLS among models, minimising the in-
fluence of inter-model differences in tropospheric chemistry on the results, and thereby (ii) isolated
differences due to transport processes. Long-term simulations, over a 20 year period (1993-2012),
were performed by each model in order to examine trends and transport-driven inter-annual vari-
ability in the stratospheric SGI of $CH Br_3$ and $CH_2Br_2$. Global monthly mean model output over
the full simulation period, along with output at a higher temporal resolution (typically hourly) over
measurement campaign periods, was requested from each group. A brief description of the models is
given in Section 2.3 and a description of the observational data used in this work is given in Section
2.4. Figure 1 summarises the approach of TransCom-VSLS and its broad objectives.

### 2.1  Tracers and oceanic emission fluxes

Owing to significant differences in the magnitude and spatial distribution of VSLS emission fluxes,
among previously published inventories (Hossaini et al., 2013), all models ran with multiple $CH Br_3$
and $CH_2Br_2$ tracers. Each of these tracers used a different set of prescribed surface emissions. Trac-
ers named "$CH Br_3\_L$", "$CH Br_3\_O$" and "$CH Br_3\_Z$" used the inventories of Liang et al. (2010),
Ordóñez et al. (2012) and Ziska et al. (2013), respectively. These three studies also reported emis-
sion fluxes for $CH_2Br_2$, and thus the same (L/O/Z) notation applies to the model $CH_2Br_2$ tracers, as
summarised in Table 1. As these inventories were recently described and compared by Hossaini et al.

(2013), only a brief description of each is given below. Surface $CHBr_3$/$CH_2Br_2$ emission maps for each inventory are given in the Supplementary Information (Figures S1 and S2).

The Liang et al. (2010) inventory is a top-down estimate of VSLS emissions based on aircraft observations, mostly concentrated around the Pacific and North America between 1996 and 2008. Measurements of $CHBr_3$ and $CH_2Br_2$ from the following National Aeronautics and Space Administration (NASA) aircraft campaigns were used to derive the ocean fluxes: PEM-Tropics, TRACE-P, INTEX, TC4, ARCTAS, STRAT, Pre-AVE and AVE. This inventory is aseasonal and assumes the

same spatial distribution of emissions for $CHBr_3$ and $CH_2Br_2$. The Ordóñez et al. (2012) inventory is also a top-down estimate based on the same set of aircraft measurements with the addition of the NASA POLARIS and SOLVE campaigns. This inventory weights tropical ($\pm20°$ latitude) $CHBr_3$ and $CH_2Br_2$ emissions according to a monthly-varying satellite climatology of chlorophyll a (chl a), a proxy for oceanic bio-productivity, providing some seasonality to the emission fluxes. The

Ziska et al. (2013) inventory is a bottom-up estimate of VSLS emissions, based on a compilation of seawater and ambient air measurements of $CHBr_3$ and $CH_2Br_2$. Climatological, aseasonal emission maps of these VSLS were calculated using the derived sea-air concentration gradients and a commonly used sea-to-air flux parameterisation; considering wind speed, sea surface temperature and salinity (Nightingale et al., 2000).

## 2.2   Tropospheric chemistry

Participating models considered chemical loss of $CHBr_3$ and $CH_2Br_2$ through oxidation by the hydroxyl radical (OH) and by photolysis. These loss processes are comparable for $CHBr_3$, with photolysis contributing $\sim60\%$ of the $CHBr_3$ chemical sink at the surface (Hossaini et al., 2010). For $CH_2Br_2$, photolysis is a minor tropospheric sink, with its loss dominated by OH-initiated oxidation.

The overall local lifetimes of $CHBr_3$ and $CH_2Br_2$ in the tropical marine boundary layer have recently been evaluated to be 15 (13-17) and 94 (84-114) days, respectively (Carpenter and Reimann, 2014). These values are calculated based on [OH] = $1\times10^6$ molecules $cm^{-3}$, T = 275 K and with a global annual mean photolysis rate. For completeness, models also considered loss of $CHBr_3$ and $CH_2Br_2$ by reaction with atomic oxygen ($O(^1D)$) and chlorine (Cl) radicals. However, these are generally

very minor loss pathways owing to the far larger relative abundance of tropospheric OH and the respective rate constants for these reactions. Kinetic data (Table 1) was taken from the most recent Jet Propulsion Laboratory (JPL) data evaluation (Sander et al., 2011). Note, the focus and design of TransCom-VSLS was to constrain the stratospheric SGI of VSLS, thus product gases - formed following the breakdown of $CHBr_3$ and $CH_2Br_2$ in the TTL (Werner et al. 2016, in prep) - and the

stratospheric PGI of bromine was not considered.

    Participating models ran with the same global monthly-mean oxidant fields. For OH, $O(^1D)$ and Cl, these fields were the same as those used in the previous TransCom-$CH_4$ model inter-comparison (Patra et al., 2011). Within the TransCom framework, these fields have been exten-

sively used and evaluated and shown to give a realistic simulation of the tropospheric burden and
lifetime of methane and also methyl chloroform. Models also ran with the same monthly-mean
$CHBr_3$ and $CH_2Br_2$ photolysis rates, calculated offline from the TOMCAT chemical transport model
(Chipperfield, 2006). TOMCAT has been used extensively to study the tropospheric chemistry of
VSLS (e.g. Hossaini et al., 2010, 2012b, 2015b) and photolysis rates from the model were used to
evaluate the lifetime of VSLS for the recent WMO Ozone Assessment Report (Carpenter and Reimann,
2014).

## 2.3 Participating models and output

Eight global models (ACTM, B3DCTM, EMAC, MOZART, NIES-TM, STAG, TOMCAT and UKCA)
and 3 of their variants (see Table 2) participated in TransCom-VSLS. All the models are offline
chemical transport models (CTMs), forced with analysed meteorology (e.g. winds and temperature
fields), with the exception of EMAC and UKCA which are free-running chemistry-climate models
(CCMs), calculating winds and temperature online. The horizontal resolution of models ranged from
$\sim 1° \times 1°$ (longitude $\times$ latitude) to $3.75° \times 2.5°$. In the vertical, the number of levels varied from 32
to 85, with various coordinate systems. A summary of the models and their salient features is given
in Table 2. Note, these features do not necessarily link to model performance as evaluated in this
work. Note also, approximately half of the models used ECMWF ERA-Interim meteorological data.
In terms of mean upwelling in the tropics, where stratospheric bromine SGI takes place, there is
generally good agreement between the most recent major reanalysis products from ECMWF, JMA
and NCEP (e.g. Harada et al., 2015). Therefore, we do not expect a particular bias in our results from
use of ERA-Interim.

Three groups, the Karlsruhe Institute of Technology (KIT), the University of Leeds (UoL) and
the University of Cambridge (UoC), submitted output from an additional set of simulations using
variants of their models. KIT ran the EMAC model twice, as a free running model (here termed
"EMAC_F") and also in *nudged* mode (EMAC_N). The UoL performed two TOMCAT simula-
tions, the first of which used the model's *standard* convection parameterisation, based on the mass
flux scheme of Tiedtke (1989). The second TOMCAT simulation ("TOMCAT_conv") used archived
convective mass fluxes, taken from the ECMWF ERA-Interim reanalysis. A description and evalua-
tion of these TOMCAT variants is given in Feng et al. (2011). In order to investigate the influence of
resolution, the UoC ran two UKCA model simulations with different horizontal/vertical resolutions.
The horizontal resolution in the "UKCA_high" simulation was a factor of 4 (2 in 2 dimensions)
greater than that of the *standard* UKCA run (Table 2).

All participating models simulated the 6 $CHBr_3$ and $CH_2Br_2$ tracers (see Section 2.1) over a 20
year period; 01/01/1993 to 31/12/2012. This period was chosen as it (i) encompasses a range of field
campaigns during which VSLS measurements were taken and (ii) allows the strong El Niño event of
1997/1998 to be investigated in the analysis of SGI trends. The monthly mean volume mixing ratio

(vmr) of each tracer was archived by each model on the same 17 pressure levels, extending from the surface to 10 hPa over the full simulation period. The models were also sampled hourly at 15 surface sites over the full simulation period and during periods of recent ship/aircraft measurement campaigns, described in Section 2.4 below. Note, the first two years of simulation were treated as spin up and output was analysed post 1995.

## 2.4 Observational data and processing

### 2.4.1 Surface

Model output was compared to and evaluated against a range of observational data. At the surface, VSLS measurements at 15 sites were considered (Table 3). All sites except one form part of the ongoing global monitoring program (see http://www.esrl.noaa.gov/gmd) of the National Oceanic and Atmospheric Administration's Earth System Research Laboratory (NOAA/ESRL). Further details related to the sampling network are given in Montzka et al. (2011). Briefly, NOAA/ESRL measurements of $CHBr_3$ and $CH_2Br_2$ are obtained from whole air samples, collected approximately weekly into paired steel or glass flasks, prior to being analysed using gas chromatography/mass spectrometry (GC/MS) in their central Boulder laboratory. Here, the climatological monthly mean mole fractions of these VSLS were calculated at each site based on monthly mean surface measurements over the 01/01/98 to 31/12/2012 period (except SUM, THD and SPO which have shorter records). Similar climatological fields of $CHBr_3$, $CH_2Br_2$ were calculated from each model's hourly output sampled at each location.

Surface measurements of $CHBr_3$ and $CH_2Br_2$, obtained by the University of Cambridge in Malaysian Borneo (Tawau, site "TAW", Table 3), were also considered. A description of these data is given in Robinson et al. (2014). Briefly, in-situ measurements were made using the $\mu$-Dirac gas chromatograph instrument with electron capture detection (GC-ECD) (e.g. Pyle et al., 2011). Measurements at TAW are for a single year (2009) only, making the observed record at this site far shorter than that at NOAA/ESRL stations discussed above.

A subset of models also provided hourly output over the period of the TransBrom and SHIVA (Stratospheric Ozone: Halogen Impacts in a Varying Atmosphere) ship cruises. During both campaigns, surface $CHBr_3$ and $CH_2Br_2$ measurements were obtained on-board the Research Vessel (R/V) *Sonne*. TransBrom sampled along a meridional transect of the West Pacific, from Japan to Australia, during October 2009 (Krüger and Quack, 2013). SHIVA was a European Union (EU)-funded project to investigate the emissions, chemistry and transport of VSLS (http://shiva.iup.uni-heidelberg.de/). Ship-borne measurements of surface $CHBr_3$ and $CH_2Br_2$ were obtained in November 2011, with sampling extending from Singapore to the Philippines, within the South China Sea and along the northern coast of Borneo (Fuhlbrügge et al., 2015). The ship track is shown in Figure 2.

 **2.4.2 Aircraft**

Observations of $CHBr_3$ and $CH_2Br_2$ from a range of aircraft campaigns were also used (Figure 2). As (i) the troposphere-to-stratosphere transport of air (and VSLS) primarily occurs in the tropics, and (ii) because VSLS emitted in the extratropics have a negligible impact on stratospheric ozone (Tegtmeier et al., 2015), TransCom-VSLS focused on aircraft measurements obtained in the latitude range 30°N to 30°S. Hourly model output was interpolated to the relevant aircraft sampling location, allowing for point-by-point model-measurement comparisons. A brief description of the aircraft campaigns follows.

The HIAPER Pole-to-Pole Observations (HIPPO) project (http://www.eol.ucar.edu/projects/hippo) comprised a series of aircraft campaigns between 2009 and 2011 (Wofsy et al., 2011), supported by the National Science Foundation (NSF). Five campaigns were conducted; HIPPO-1 (January 2009), HIPPO-2 (November 2009), HIPPO-3 (March/April 2010), HIPPO-4 (June 2011) and HIPPO-5 (August/September 2011). Sampling spanned a range of latitudes, from near the North Pole to coastal Antarctica, on board the NSF Gulfstream V aircraft, and from the surface to ~14 km over the Pacific Basin. Whole air samples, collected in stainless steel and glass flasks, were analysed by two different laboratories using GC/MS; NOAA/ESRL and the University of Miami. HIPPO results from both laboratories are provided on a scale consistent with NOAA/ESRL.

The SHIVA aircraft campaign, based in Miri (Malaysian Borneo), was conducted during November—December 2011. Measurements of $CHBr_3$ and $CH_2Br_2$ were obtained during 14 flights of the DLR Falcon aircraft, with sampling over much of the northern coast of Borneo, within the South China and Sulu seas, up to an altitude of ~12 km (Sala et al., 2014; Fuhlbrügge et al., 2015). VSLS measurements were obtained by two groups; the University of Frankfurt (UoF) and the University of East Anglia (UEA). UoF measurements were made using an in-situ GC/MS system (Sala et al., 2014), while UEA analysed collected whole air samples, using GC/MS.

CAST (Coordinated Airborne Studies in the Tropics) is an ongoing research project funded by the UK Natural Environment Research Council (NERC) and is a collaborative initiative with the NASA ATTREX programme (see below). The CAST aircraft campaign, based in Guam, was conducted in January-February 2014 with VSLS measurements made by the University of York on-board the FAAM (Facility for Airborne Atmospheric Measurements) BAe-146 aircraft, up to an altitude of ~8 km. These observations were made by GC/MS collected from whole air samples as described in Andrews et al. (2016).

Observations of $CHBr_3$ and $CH_2Br_2$ within the TTL and lower stratosphere (up to ~20 km) were obtained during the NASA (i) Pre-Aura Validation Experiment (Pre-AVE), (ii) Costa Rica Aura Validation Experiment (CR-AVE) and (iii) Airborne Tropical TRopopause EXperiment (ATTREX) missions. The Pre-AVE mission was conducted in 2004 (January-February), with measurements obtained over the equatorial eastern Pacific during 8 flights of the high altitude WB-57 aircraft. The CR-AVE mission took place in 2006 (January-February) and sampled a similar region around

Costa Rica (Figure 2), also with the WB-57 aircraft (15 flights). The ATTREX mission consists of an ongoing series of aircraft campaigns using the unmanned Global Hawk aircraft. Here, $CHBr_3$ and $CH_2Br_2$ measurements from 10 flights of the Global Hawk, over two ATTREX campaigns, were used. The first campaign (February-March, 2013) sampled large stretches of the north east and central Pacific ocean, while the second campaign (January-March, 2014) sampled predominantly the West Pacific, around Guam. During Pre-AVE, CR-AVE and ATTREX, VSLS measurements were obtained by the University of Miami following GC/MS analysis of collected whole air samples.

## 3    Results and Discussion

### 3.1    Model-observation comparisons: surface

In this section, we evaluate the models in terms of (i) their ability to capture the observed seasonal cycle of $CHBr_3$ and $CH_2Br_2$ at the surface and (ii) the absolute agreement to the observations. We focus on investigating the relative performance of each of the tested emission inventories, within a given model, and the performance of the inventories across the ensemble.

#### 3.1.1    Seasonality

We first consider the seasonal cycle of $CHBr_3$ and $CH_2Br_2$ at the locations given in Table 3. Figure 3 compares observed and simulated ($CHBr_3$_L tracer) monthly mean anomalies, calculated by subtracting the climatological monthly mean $CHBr_3$ surface mole fraction from the climatological annual mean (to focus on the seasonal variability). Based on photochemistry alone, in the northern hemisphere (NH) one would expect a $CHBr_3$ winter (Dec-Feb) maximum owing to a reduced chemical sink (e.g. slower photolysis rates and lower [OH]) and thereby a relatively longer $CHBr_3$ lifetime. This seasonality, apparent at most NH sites shown in Figure 3, is particularly pronounced at high-latitudes ($>60°$N, e.g. ALT, BRW and SUM), where the amplitude of the observed seasonal cycle is greatest. A number of features are apparent from these comparisons. First, in general most models reproduce the observed phase of the $CHBr_3$ seasonal cycle well, even with emissions that do not vary seasonally, suggesting that seasonal variations in the $CHBr_3$ chemical sink are generally well represented. For example, model-measurement correlation coefficients ($r$), summarised in Table 4, are $>0.7$ for at least 80% of the models at 7 of 11 NH sites. Second, at some sites, notably MHD, THD, CGO and PSA, the observed seasonal cycle of $CHBr_3$ is not captured well by virtually all of the models (see discussion below). Third, at most sites the amplitude of the seasonal cycle is generally consistent across the models (within a few percent, excluding clear outliers). The cause of outliers at a given site are likely in part related to the model sampling error, including distance of a model grid from the measurement site and resolution (as was shown for $CO_2$ in Patra et al. (2008)). These instances are rare for VSLS but can be seen in B3DCTM's output in Figure 3 for $CHBr_3$ at SMO. B3DCTM ran at a relatively coarse horizontal resolution ($3.75°$) and with less vertical layers

(40) compared to most other models. Note, it also has the simplest implementation of boundary layer
mixing (Table 2). The above behaviour is also seen at SMO but to a lesser extent for $CH_2Br_2$, for
which the seasonal cycle is smaller (see below). The STAG model also produces distinctly different
features in the seasonal cycle of both species at some sites (prominently at CGO, SMO and HFM).
We attribute these deviations to STAG's parameterisation of boundary layer mixing, noting that dif-
ferences for $CHBr_3$ are greater at KUM than at MLO – two sites in very close proximity but with
the latter elevated at $\sim$3000 metres above sea level (i.e. above the boundary layer). With respect to
the observations, the amplitude of the seasonal cycle is either under- (e.g. BRW) or over-estimated
(e.g. KUM) at some locations, by all of the models. This possibly reflects a more systematic bias in
the prescribed $CHBr_3$ loss rate and/or relates to emissions, though this effect is generally small and
localised.

A similar analysis has been performed to examine the seasonal cycle of surface $CH_2Br_2$. Ob-
served and simulated monthly mean anomalies, calculated in the same fashion as those for $CHBr_3$
above, are shown in Figure 4 and correlation coefficients are given in Table 5. The dominant chem-
335 ical sink of $CH_2Br_2$ is through OH-initiated oxidation and thus its seasonal cycle at most stations
reflects seasonal variation in [OH] and temperature. At most sites, this gives rise to a minimum in
the surface mole fraction of $CH_2Br_2$ during summer months, owing to greater [OH] and tempera-
ture, and thereby a faster chemical sink. Relative to $CHBr_3$, $CH_2Br_2$ is considerably longer-lived
(and thus well mixed) near the surface, meaning the amplitude of the seasonal cycle is far smaller.
At most sites, most models capture the observed phase and amplitude of the $CH_2Br_2$ seasonal cy-
cle well, though as was the case for $CHBr_3$, agreement in the southern hemisphere (SH, e.g. SMO,
CGO, PSA) seems poorest. For example, at SMO and CGO only 40% of the models are positively
correlated to the observations with $r > 0.5$ (Table 5). The NIES-TM model does not show major
differences from other models for $CHBr_3$, but outliers for $CH_2Cl_2$ at SH sites (SMO to SPO) are
345 apparent. We were unable to assign any specific reason for the inter-species differences seen for this
model.

At two sites (MHD and THD) almost none of the models reproduce the observed $CHBr_3$ seasonal
cycle, exhibiting an anti-correlation with the observed cycle (see bold entries in Table 4). Here, the
simulated cycle follows that expected from seasonality in the chemical sink. At MHD, seasonality
in the local emission flux is suggested to be the dominant factor controlling the seasonal cycle of
surface $CHBr_3$ (Carpenter et al., 2005). This leads to the observed summer maximum (as shown
in Figure 3) and is not represented in the models' $CHBr_3\_L$ tracer which, at the surface, is driven
by the aseasonal emission inventory of Liang et al. (2010). A similar summer maximum seasonal
cycle is observed for $CH_2Br_2$, also not captured by the models' $CH_2Br_2\_L$ tracer. To investigate the
355 sensitivity of the model-measurement correlation to the prescribed surface fluxes, multi-model mean
(MMM) surface $CHBr_3$ and $CH_2Br_2$ fields were calculated for each tracer (i.e. for each emission
inventory considered) and each site. Figure 5 shows calculated MMM $r$ values at each site for $CHBr_3$

and $CH_2Br_2$. For $CHBr_3$, $r$ generally has a low sensitivity to the choice of emission fluxes at most sites (e.g. ALT, SUM, BRW, LEF, NWR, KUM, MLO, SPO), though notably at MHD, use of the Ziska et al. (2013) inventory (which is aseasonal) reverses the sign of $r$ to give a strong positive correlation (MMM $r > 0.70$) against the observations. Individual model $r$ values for MHD are given in Table S1 of the Supplementary Information. With the exception of TOMCAT, TOMCAT_CONV and UKCA_HI, the remaining 7 models each reproduce the MHD $CHBr_3$ seasonality well (with $r > 0.65$). That good agreement is obtained with the Ziska aseasonal inventory, compared to the other aseasonal inventories considered, highlights the importance of the $CHBr_3$ emission distribution, with respect to transport processes, serving this location. We suggest that the summertime transport of air that has experienced relatively large $CHBr_3$ emissions north/north-west of MHD is the cause of the apparent seasonal cycle seen in most models using the Ziska inventory (example animations of the seasonal evolution of surface $CHBr_3$ are given in the Supplementary Information to visualise this). Note also, the far better absolute model-measurement agreement obtained at MHD for models using this inventory (Supplementary Figure S3). At other sites, such as TAW, no clear seasonality is apparent in the observed background mixing ratios of $CHBr_3$ and $CH_2Br_2$ (Robinson et al., 2014). Here, the models exhibit little or no significant correlation to measured values and are unlikely to capture small-scale features in the emission distribution (e.g the contribution from local aquaculture) that conceivably contribute to observed levels of $CHBr_3$ and $CH_2Br_2$ in this region (Robinson et al., 2014).

### 3.1.2 Absolute agreement

To compare the absolute agreement between a model (M) and an observation (O) value, for each monthly mean surface model-measurement comparison, the mean absolute percentage error (MAPE, equation 1) was calculated for each model tracer. Figure 6 shows the $CHBr_3$ and $CH_2Br_2$ tracer that provides the lowest MAPE (i.e. best agreement) for each model (indicated by the fill colour of cells). The numbers within the cells give the MAPE value itself, and therefore correspond to the "best agreement" that can be obtained from the various tracers with the emission inventories that were tested.

$$\text{MAPE} = \frac{100}{n} \sum_{t=1}^{n} \left| \frac{M_t - O_t}{O_t} \right| \tag{1}$$

For both $CHBr_3$ and $CH_2Br_2$, within any given model, no single emission inventory is able to provide the best agreement at all surface locations (i.e. from the columns in Figure 6). This was previously noted by Hossaini et al. (2013) using the TOMCAT model, and to some degree likely reflects the geographical coverage of the observations used to create the emission inventories. Hossaini et al. (2013) also noted significant differences between simulated and observed $CHBr_3$ and $CH_2Br_2$, using

the same inventory; i.e. at a given location, low CHBr$_3$ MAPE (good agreement) does not necessarily accompany a corresponding low CH$_2$Br$_2$ MAPE using the same inventory.

A key finding of this study is that significant inter-model differences are also apparent (i.e. see rows in Figure 6 grid). For example, for CHBr$_3$, no single inventory performs best across the full range of models at any given surface site. TOMCAT and B3DCTM - both of which are driven by ERA-Interim - agree on the best CHBr$_3$ inventory (lowest MAPE) at approximately half of the 17 sites considered. This analysis implies that, on a global scale, the "performance" of emission inventories is somewhat model-specific and highlights the challenges of evaluating such inventories. Previous conclusions as to the *best* performing VSLS inventories, based on single model simulations (Hossaini et al., 2013), must therefore be treated with caution. When one considers that previous modelling studies (Warwick et al., 2006; Liang et al., 2010; Ordóñez et al., 2012), each having derived different VSLS emissions based on aircraft observations, and having different tropospheric chemistry, report generally good agreement between their respective model and observations, our findings are perhaps not unexpected. However, we note also that few VSLS modelling studies have used long-term surface observations to evaluate their models, as performed here. This suggests any attempts to reconcile estimates of global VSLS emissions, obtained from different modelling studies, need to consider the influence of inter-model differences.

As the chemical sink of VSLS was consistent across all models, the inter-model differences discussed above are attributed primarily to differences in the treatment and implementation of transport processes. This includes convection and boundary layer mixing, both of which can significantly influence the near-surface abundance of VSLS in the real (Fuhlbrügge et al., 2013, 2015) and model (Zhang et al., 2008; Feng et al., 2011; Hoyle et al., 2011) atmospheres, and are parameterised in different ways (Table 2). On this basis, it is not surprising that different CTM setups lead to differences in the surface distribution of VSLS, nor that differences are apparent between CTMs that use the same meteorological input fields. Indeed, such effects have also been observed in previous model intercomparisons (Hoyle et al., 2011). Large-scale vertical advection, the native grid of a model and its horizontal/vertical resolution may also be contributing factors, though quantifying their relative influence was beyond the scope of TransCom-VSLS. At some sites, differences among emission inventory performance are apparent between model variants that, besides transport, are otherwise identical; i.e. TOMCAT and TOMCAT_CONV entries of Figure 6.

Despite the inter-model differences in the performance of emission inventories, some generally consistent features are found across the ensemble. First, for CHBr$_3$ the tropical MAPE (see Figure 7), based on the model-measurement comparisons in the latitude range $\pm 20°$, is lowest when using the emission inventory of Ziska et al. (2013), for most (8 out of 11, $\sim$70%) of the models. This is significant as troposphere-to-stratosphere transport primarily occurs in the tropics and the Ziska et al. (2013) inventory has the lowest CHBr$_3$ emission flux in this region (and globally, Table 1). Second, for CH$_2$Br$_2$, the tropical MAPE is lowest for most (also $\sim$70%) of the models

when using the Liang et al. (2010) inventory, which also has the lowest global flux of the three inventories tested. For a number of models, a similar agreement is also obtained with Ordóñez et al. (2012) inventory, as the two are broadly similar in magnitude/distribution (Hossaini et al., 2013). For $CH_2Br_2$, the Ziska et al. (2013) inventory performs poorest across the ensemble (models generally overestimate $CH_2Br_2$ with this inventory). Overall, the tropical MAPE for a given model is more sensitive to choice of emission inventory for $CHBr_3$ than $CH_2Br_2$ (Figure 7). Based on each model's *preferred* inventory (i.e. from Figure 7), the tropical MAPE is generally ∼40% for $CHBr_3$ and <20% for $CH_2Br_2$ (in most models). One model (STAG) exhibited a MAPE of >50% for both species, regardless of the choice of emission inventory, and was therefore omitted from the subsequent model-measurement comparisons to aircraft data and also from the multi-model mean SGI estimate derived in Section 3.5.

For the 5 models that submitted hourly output over the period of the SHIVA (2011) and TransBrom (2009) ship cruises, Figures 8 and 9 compare the multi-model mean (MMM) $CHBr_3$ and $CH_2Br_2$ mixing ratio (and the model spread) to the observed values. Note, the MMM was calculated based on each model's preferred tracer (i.e. preferred emissions inventory). Generally, the models reproduce the observed mixing ratios from SHIVA well, with a MMM campaign MAPE of 25% or less for both VSLS. This is encouraging as SHIVA sampled in the tropical West Pacific region, where rapid troposphere-to-stratosphere transport of VSLS likely occurs (e.g. Aschmann et al., 2009; Liang et al., 2014) and where VSLS emissions, weighted by their ozone depletion potential, are largest (Tegtmeier et al., 2015). Model-measurement comparisons during TransBrom are varied with models generally underestimating observed $CHBr_3$ and $CH_2Br_2$ during significant portions of the cruise. The underestimate is most pronounced close to the start and end of the cruise during which observed mixing ratios were more likely influenced by coastal emissions, potentially underestimated in global-scale models. Note, TransBrom also sampled sub-tropical latitudes (see Figure 2).

Overall, our results show that most models capture the observed seasonal cycle and the magnitude of surface $CHBr_3$ and $CH_2Br_2$ reasonably well, using a combination of emission inventories. Generally, this leads to a realistic surface distribution at most locations, and thereby provides good agreement between models and aircraft observations above the boundary layer; see Section 3.2 below.

## 3.2   Model-observation comparisons: free troposphere

We now evaluate modelled profiles of $CHBr_3$ and $CH_2Br_2$ using observations from a range of recent aircraft campaigns (see Section 2.4). Note, for these comparisons, and from herein unless noted, all analysis is performed using the *preferred* $CHBr_3$ and $CH_2Br_2$ tracer for each model (i.e. preferred emissions inventory), as was diagnosed in the previous discussion (i.e. from Figure 7, see also Section 3.1.2). This approach ensures that an estimate of stratospheric bromine SGI, from a given

model, is based on a simulation in which the optimal $CHBr_3/CH_2Br_2$ model-measurement agree-
ment at the surface was acheived. The objective of the comparisons below is to show that the models
produce a realistic simulation of $CHBr_3$ and $CH_2Br_2$ in the tropical free troposphere and to test
model transport of $CHBr_3$ and $CH_2Br_2$ from the surface to high altitudes, against that from atmo-
spheric measurements. Intricacies of individual model-measurement comparison are not discussed.
Rather, Figure 10 compares MMM profiles (and the model spread) of $CHBr_3$ and $CH_2Br_2$ mixing
ratio to observed campaign means within the tropics ($\pm20°$ latitude). Generally model-measurement
agreement, diagnosed by both the campaign-averaged MAPE and the correlation coefficient ($r$) is
excellent during most campaigns. For all of the 7 campaigns considered, the modelled MAPE for
$CHBr_3$ is $\leq35\%$ ($\leq20\%$ for $CH_2Br_2$). The models also capture much of the observed variability
throughout the observed profiles, including, for example, the signature "c-shape" of convection in
the measured $CHBr_3$ profile from SHIVA and HIPPO-1 (panel (a), 2nd and 3rd rows of Figure 10).
Correlation coefficients between modelled and observed $CHBr_3$ are $\geq0.8$ for 5 of the 7 campaigns
and for $CH_2Br_2$ are generally $>0.5$.

It is unclear why model-measurement agreement (particularly the $CHBr_3$ MAPE) is poorest for
the HIPPO-4 and HIPPO-5 campaigns. However, we note that at most levels MMM $CHBr_3$ and
$CH_2Br_2$ falls within $\pm1$ standard deviation ($\sigma$) of the observed mean. Note, an underestimate of
surface $CHBr_3$ does not generally translate to a consistent underestimate of measured $CHBr_3$ at
higher altitude. Critically, for the most part, the models are able to reproduce observed values of
both gases well at $\sim$12-14 km, within the lower TTL. Recall that the TTL is defined as the layer
between the level of main convective outflow ($\sim$200 hPa, $\sim$12 km) and the tropical tropopause
($\sim$100 hPa, $\sim$17 km) (Gettelman and Forster, 2002). For a given model, simulations using the non-
preferred tracers (i.e. with different $CHBr_3/CH_2Br_2$ emission inventories, not shown), generally lead
to worse model-measurement agreement in the TTL. This is not surprising as model-measurement
agreement at the surface is poorer in those simulations (as discussed in Section 3.1.2.).

Overall, given the large spatial/temporal variability in observed VSLS mixing ratios, in part due
to the influence of transport processes, global-scale models driven by aseasonal emissions and using
parameterised sub-grid scale transport schemes face challenges in reproducing VSLS observations
in the tropical atmosphere. Yet despite this, we find that the TransCom-VSLS models generally
provide a very good simulation of the tropospheric abundance of $CHBr_3$ and $CH_2Br_2$, particularly
in the important tropical West Pacific region (e.g. SHIVA comparisons).

## 3.3 Model-observation comparisons: TTL and lower stratosphere

Figure 11 compares model profiles of $CHBr_3$ and $CH_2Br_2$ with high altitude measurements obtained
in the TTL, extending into the tropical lower stratosphere. Across the ensemble, model-measurement
agreement is varied but generally the models capture observed $CHBr_3$ from the Pre-AVE and CR-
AVE campaigns, in the Eastern Pacific, well. It should be noted that the number of observations

500 varies significantly between these two campaigns; CR-AVE had almost twice the number of flights as Pre-AVE and this is reflected in the larger variability in the observed profile, particularly in the lower TTL. For both campaigns, the models capture the observed gradients in $CHBr_3$ and variability throughout the profiles; model-measurement correlation coefficients ($r$) for all of the models are >0.93 and >0.88 for Pre-AVE and CR-AVE, respectively. In terms of absolute agreement, 100% of 505 the models fall within $\pm 1\sigma$ of the observed $CHBr_3$ mean at the tropopause during Pre-AVE (and $\pm 2\sigma$ for CR-AVE). For both campaigns, virtually all models are within the measured (min-max) range (not shown) around the tropopause.

During both ATTREX campaigns, larger $CHBr_3$ mixing ratios were observed in the TTL (panels c and d of Figure 11). This reflects the location of the ATTREX campaigns compared to Pre-AVE and 510 CR-AVE; over the tropical West Pacific, the level of main convective outflow extends deeper into the TTL compared to the East Pacific (Gettelman and Forster, 2002), allowing a larger portion of the surface $CHBr_3$ mixing ratio to detrain at higher altitudes. Overall, model-measurement agreement of $CHBr_3$ in the TTL is poorer during the ATTREX campaigns, with most models exhibiting a low bias between 14-16 km altitude. MOZART and UKCA simulations (which prefer the Liang $CHBr_3$ 515 inventory) exhibit larger mixing ratios in the TTL, though are generally consistent with other models around the tropopause. Most ($\geq 70\%$) of the models reproduce $CHBr_3$ at the tropopause to within $\pm 1\sigma$ of the observed mean and all the models are within the measured range (not shown) during both ATTREX campaigns. Model-measurement $CHBr_3$ correlation is >0.8 for each ATTREX campaign, showing that again much of the observed variability throughout the $CHBr_3$ profiles is captured. The 520 same is true for $CH_2Br_2$, with $r$ >0.84 for all but one of the models during Pre-AVE and $r$ >0.88 for all of the models in each of the other campaigns.

Overall, mean $CHBr_3$ and $CH_2Br_2$ mixing ratios around the tropopause, observed during the 2013/2014 ATTREX missions, are larger than the mean mixing ratios (from previous aircraft campaigns) reported in the latest WMO Ozone Assessment Report (Table 1-7 of Carpenter and Reimann 525 (2014)). As noted, this likely reflects the location at which the measurements were made; ATTREX 2013/2014 sampled in the tropical West and Central Pacific, whereas the WMO estimate is based on a compilation of measurements with a paucity in that region. From Figure 11, observed $CHBr_3$ and $CH_2Br_2$ at the tropopause was (on average) ~0.35 ppt and ~0.8 ppt, respectively, during ATTREX 2013/2014, compared to the 0.08 (0.00—0.31) ppt $CHBr_3$ and 0.52 (0.3-–0.86) ppt $CH_2Br_2$ ranges 530 reported by Carpenter and Reimann (2014).

**3.4 Seasonal and zonal variations in the troposphere-to-stratosphere transport of VSLS**

In this section we examine seasonal and zonal variability in the loading of $CHBr_3$ and $CH_2Br_2$ in the TTL and lower stratosphere, indicative of transport processes. In the tropics, a number of previous studies have shown a marked seasonality in convective outflow around the tropopause, 535 owing to seasonal variations in convective cloud top heights (e.g. Folkins et al., 2006; Hosking et al.,

2010; Bergman et al., 2012). Such variations influence the near-tropopause abundance of brominated VSLS (Hoyle et al., 2011; Liang et al., 2014) and other tracers, such as CO (Folkins et al., 2006).

Figures 12 and 13 show the simulated seasonal cycle of $CHBr_3$ and $CH_2Br_2$, respectively, at the base of the TTL and the cold point tropopause (CPT). $CHBr_3$ exhibits a pronounced seasonal cycle at the CPT, with virtually all models showing the same phase; with respect to the annual mean and integrated over the tropics, $CHBr_3$ is most elevated during boreal winter (DJF). The amplitude of the cycle varies considerably between models, with departures from the annual mean ranging from around $\pm10\%$ to $\pm40\%$, in a given month (panel b of Figure 12). Owing to its relatively long tropospheric lifetime, particularly in the TTL ($>1$ year) (Hossaini et al., 2010), $CH_2Br_2$ exhibits a weak seasonal cycle at the CPT as it is less influenced by seasonal variations in transport.

Panels (c) and (d) of Figures 12 and 13, also show the modelled absolute mixing ratios of $CHBr_3$ and $CH_2Br_2$ at the TTL base and CPT. Annually averaged, for $CHBr_3$, the model spread results in a factor of $\sim3$ difference in simulated $CHBr_3$ at both levels (similarly, for $CH_2Br_2$ a factor of 1.5). The modelled mixing ratios fall within the measurement-derived range reported by Carpenter and Reimann (2014). The MMM $CHBr_3$ mixing ratio at the TTL base is 0.51 ppt, within the 0.2-1.1 ppt measurement-derived range. At the CPT, the MMM $CHBr_3$ mixing ratio is 0.20 ppt, also within the measured range of 0.0-0.31 ppt. On average, the models suggest a $\sim60\%$ gradient in $CHBr_3$ between the TTL base and tropopause. Similarly, the annual MMM $CH_2Br_2$ mixing ratio is 0.82 ppt at the TTL base, within the measured range of 0.6-1.2 ppt, and at the CPT is 0.73 ppt, within the measured range of 0.3-0.86 ppt. On average, the models show a $CH_2Br_2$ gradient of 10% between the two levels. These model absolute values are annual means over the whole tropical domain. However, zonal variability in VSLS loading within the TTL is expected to be large (e.g. Aschmann et al., 2009; Liang et al., 2014), owing to inhomogeneity in the spatial distribution of convection and oceanic emissions. The Indian Ocean, the Maritime Continent (incorporating Malaysia, Indonesia, and the surrounding islands and ocean), central America, and central Africa are all convectively-active regions, shown to experience particularly deep convective events with the potential, therefore, to rapidly loft VSLS from the surface into the TTL (e.g. Gettelman et al., 2002, 2009; Hosking et al., 2010). As previously noted, the absolute values can vary, though generally the TransCom-VSLS models agree on the locations with the highest VSLS mixing ratios, as seen from the zonal $CHBr_3$ anomalies at the CPT shown in Figure 14. These regions are consistent with the convective source regions discussed above. The largest $CHBr_3$ mixing ratios at the CPT are predicted over the tropical West Pacific ($20°S-20°N$, $100°E-180°E$), particularly during DJF. Integrated over the tropical domain, this signal exerts the largest influence on the $CHBr_3$ seasonal cycle at the CPT. This result is consistent with the model intercomparison of Hoyle et al. (2011), who examined the seasonal cycle of idealised VSLS-like tracers around the tropopause, and reported a similar seasonality.

While meridionally, the width of elevated $CHBr_3$ mixing ratios during DJF is similar across the models, differences during boreal summer (JJA) are apparent, particularly in the vicinity of the Asian

Monsoon (5°N-35°N, 60°E-120°E). Note, the CHBr$_3$ anomalies shown in Figure 14 correspond to departures from the mean calculated in the latitude range of $\pm 30°$, and therefore encompass most of the Monsoon region. A number of studies have highlighted (i) the role of the Monsoon in transporting pollution from east Asia into the stratosphere (e.g. Randel et al., 2010) and (ii) its potential role in the troposphere-to-stratosphere transport of aerosol precursors, such as volcanic SO$_2$ (e.g. Bourassa et al., 2012; Fromm et al., 2014). For VSLS, and other short-lived tracers, the Monsoon may also represent a significant pathway for transport to the stratosphere (e.g. Vogel et al., 2014; Orbe et al., 2015; Tissier and Legras , 2016). Here, a number of models show elevated CHBr$_3$ in the lower stratosphere over the Monsoon region, though the importance of the Monsoon with respect to the tropics as a whole varies substantially between the models. For example, from Figure 14, models such as ACTM and UKCA show far greater enhancement in CHBr$_3$ associated with the Monsoon during JJA, compared to others (e.g. MOZART, TOMCAT). A comparison of CHBr$_3$ anomalies at 100 hPa but confined to the Monsoon region, as shown in Figure 15, reveals a Monsoon signal in most of the models, but as noted above the strength of this signal varies considerably. The STAG model, which does not include a treatment of *deep* convection and has been shown to have weak ventilation through the boundary layer (Law et al., 2008), exhibits virtually no CHBr$_3$ enhancement over the Monsoon region.

The high altitude model-model differences in CHBr$_3$, highlighted in Figures 14 and 15, are attributed predominately to differences in the treatment of convection. Previous studies have shown that (i) convective updraft mass fluxes, including the vertical extent of deep convection (relevant for bromine SGI from VSLS), vary significantly depending on the implementation of convection in a given model (e.g. Feng et al., 2011) and (ii) that significantly different short-lived tracer distributions are predicted from different models using different convective parameterisations (e.g. Hoyle et al., 2011). Such parameterisations are often complex, relying on assumptions regarding detrainment levels, trigger thresholds for shallow, mid-level and/or deep convection, and vary in their approach to computing updraft (and downdraft) mass fluxes. Furthermore, the vertical transport of model tracers is also sensitive to interactions of the convective parameterisation with the boundary layer mixing scheme (also parameterised) (Rybka and Tost, 2014). On the above basis and considering that the TransCom-VSLS models implement these processes in different ways (Table 2), it was not possible to detangle transport effects within the scope of this project. However, no systematic similarities/differences between models according to input meteorology were apparent. Examining the difference between UKCA_HI and UKCA_LO reveals that horizontal resolution is a significant factor. The UKCA_HI simulation shows a greater role of the Monsoon region, likely due to differences in the distribution of surface emissions (e.g. along longer coastlines in the higher resolution model) with respect to the occurrence of convection, as shown by Russo et al. (2015). Overall, aircraft VSLS observations within this poorly sampled region are required in order to elucidate further the role of the Monsoon in the troposphere-to-stratosphere transport of brominated VSLS.

## 3.5 Stratospheric source gas injection of bromine and trends

In this section we quantify the climatological SGI of bromine from $CHBr_3$ and $CH_2Br_2$ to the tropical LS and examine inter-annual variability. The current measurement-derived range of bromine SGI ($[3\times CHBr_3] + [2\times CH_2Br_2]$ at the tropical tropopause) from these two VSLS is 1.28 (0.6-2.65) ppt Br, i.e. uncertain by a factor of $\sim$4.5 (Carpenter and Reimann, 2014). This uncertainty dominates the overall uncertainty on the *total* stratospheric bromine SGI range (0.7-3.4 ppt Br), which includes relatively minor contributions from other VSLS (e.g. $CHBr_2Cl$, $CH_2BrCl$ and $CHBrCl_2$). Given that SGI may account for up to 76% of stratospheric $Br_y^{VSLS}$ (Carpenter and Reimann, 2014) (note, $Br_y^{VSLS}$ also includes the contribution of product gas injection), constraining the contribution from $CHBr_3$ and $CH_2Br_2$ is, therefore, desirable.

The TransCom-VSLS climatological MMM estimate of Br SGI from $CHBr_3$ and $CH_2Br_2$ is 2.0 (1.2-2.5) ppt Br, with the reported uncertainty from the model spread. $CH_2Br_2$ accounts for $\sim$72% of this total, in good agreement with the $\sim$80% reported by Carpenter and Reimann (2014). The model spread encompasses the best estimate reported by Carpenter and Reimann (2014), though our best estimate is 0.72 ppt (57%) larger. The spread in the TransCom-VSLS models is also 37% lower than the Carpenter and Reimann (2014) range, suggesting that their measurement-derived range in bromine SGI from $CHBr_3$ and $CH_2Br_2$ is possibly too conservative, particularly at the lower limit (Figure 16), and from a climatological perspective. We note that (i) the TransCom-VSLS estimate is based on models, shown here, to simulate the surface to tropopause abundance of $CHBr_3$ and $CH_2Br_2$ well and (ii) represents a climatological estimate over the simulation period, 1995-2012. The measurement-derived best estimate and range (i.e. that from Carpenter and Reimann (2014)) does not include the high altitude observations over the tropical West Pacific obtained during the most recent NASA ATTREX missions. As noted in Section 3.3, mean $CHBr_3$ and $CH_2Br_2$ measured around the tropopause during ATTREX (2013/2014 missions), is at the upper end of the compilation of observed values given in the recent WMO Ozone Assessment Report (Table 1-7 of Carpenter and Reimann (2014)). Inclusion of these data would bring the WMO SGI estimate from $CHBr_3$ and $CH_2Br_2$ closer to the TransCom-VSLS estimate reported here. For context, the TransCom-VSLS MMM estimate of Br SGI from $CHBr_3$ and $CH_2Br_2$ (2.0 ppt Br) represents 10% of *total* stratospheric $Br_y$ (i.e. considering long-lived sources gases also) - estimated at $\sim$20 ppt in 2011 (Carpenter and Reimann, 2014).

The TransCom-VSLS MMM SGI range discussed above is from $CHBr_3$ and $CH_2Br_2$ only. *Minor* VSLS, including $CHBr_2Cl$, $CH_2BrCl$, $CHBrCl_2$, $C_2H_5Br$, $C_2H_4Br$ and $C_3H_7Br$, are estimated to contribute a further 0.08 to 0.71 ppt Br through SGI (Carpenter and Reimann, 2014). If we add this contribution on to our MMM estimate of bromine SGI from $CHBr_3$ and $CH_2Br_2$, a reasonable estimate of 1.28 to 3.21 ppt Br is derived from our results for the *total* SGI range. This range is 28% smaller than the equivalent estimate of total SGI reported by Carpenter and Reimann (2014), because of the constraint on the contribution from $CHBr_3$ and $CH_2Br_2$, as discussed above.

Our uncertainty estimate on simulated bromine SGI (from the model spread) reflects inter-model variability, primarily due to differences in transport, but does not account for uncertainty on the chemical factors influencing the loss rate and lifetime of VSLS (e.g. tropospheric [OH]) - as all of the models used the same prescribed oxidants. However, Aschmann and Sinnhuber (2013) found that the stratospheric SGI of Br exhibited a low sensitivity to large perturbations to the chemical loss rate of $CHBr_3$ and $CH_2Br_2$; a $\pm50\%$ perturbation to the loss rate changed bromine SGI by 2% at most in their model sensitivity experiments. Furthermore, our SGI range is compatible with recent model SGI estimates that used different [OH] fields; for example, Fernandez et al. (2014) simulated a stratospheric SGI of 1.7 ppt Br from $CHBr_3$ and $CH_2Br_2$.

We found no clear long-term transport-driven trend in the stratospheric SGI of bromine. Clearly, this result is limited to the study period examined and does not preclude potential future changes due to climate change, as suggested by some studies (e.g. Hossaini et al., 2012b). In terms of inter-annual variability, the simulated annual mean bromine SGI varied by $\pm5\%$ around the climatological mean (panel (b) of Figure 16) over the simulation period (small in the context of total stratospheric $Br_y$, see above). Naturally, this encompasses inter-annual variability of both $CHBr_3$ and $CH_2Br_2$ reaching the tropical LS. The latter of which is far smaller and given that $CH_2Br_2$ is the larger contributor to SGI, dampens the overall inter-annual variability. Note, inter-annual changes in emissions, [OH] or photolysis rates were not quantified here (only transport). On a monthly basis, the amount of $CHBr_3$ reaching the tropical LS can clearly exhibit larger variability. $CHBr_3$ anomalies (calculated as monthly departures from the climatological monthly mean mixing ratio) at the tropical tropopause are shown in Figure 17. Also shown in Figure 17 is the Multivariate ENSO Index (MEI) - a time-series which characterises ENSO intensity based on a range of meteorological and oceanographic components (Wolter and Timlin, 1998). See also: http://www.esrl.noaa.gov/psd/enso/mei/. The transport of $CHBr_3$ (and $CH_2Br_2$, not shown) to the tropical LS is strongly correlated ($r$ values ranging from 0.6 to 0.75 across the ensemble) to ENSO activity over the Eastern Pacific (owing to the influence of sea surface temperature on convection). For example, a clear signal of the very strong El Niño event of 1997/1998 is apparent in the models (i.e. with enhanced $CHBr_3$ at the tropopause) supporting the notion that bromine SGI is sensitive to such climate modes, in *this* region (Aschmann et al., 2011). However, when averaged over the tropics no strong correlation between VSLS loading in the LS and the MEI (or just sea surface temperature) was found across the ensemble. We suggest that zonal variations in SST anomalies (and convective activity) associated with ENSO, with warming in some regions and cooling in others, has a cancelling effect on the tropical mean bromine SGI. Indeed, previous model studies have showed a marked zonal structure in $CHBr_3$/$CH_2Br_2$ loading in the LS in strong ENSO years, with relative increases and decreases with respect to climatological averages depending on region (Aschmann et al., 2011). Further investigation, beyond the scope of this work, is needed to determine the sensitivity of *total* stratospheric

$Br_y^{VSLS}$ (i.e. including the contribution from product gas injection), to this and other modes of climate variability.

**4  Summary and Conclusions**

Understanding the chemical and dynamical processes which influence the atmospheric loading of VSLS in the present, and how these processes may change in the future, is important to understand the role of VSLS in a number of issues. In the context of the stratosphere, it is important to (i) determine the relevance of VSLS for assessments of $O_3$ layer recovery timescales (Yang et al.,
2014), (ii) assess the full impact of proposed stratospheric geoengineering strategies (Tilmes et al., 2012) and (iii) accurately quantify the ozone-driven radiative forcing (RF) of climate (Hossaini et al., 2015a). Here we performed the first concerted multi-model intercomparison of halogenated VSLS. The overarching objective of TransCom-VSLS was to provide a reconciled model estimate of the SGI of bromine from $CHBr_3$ and $CH_2Br_2$ to the lower stratosphere and to investigate inter-model
differences due to emissions and transport processes. Participating models performed simulations over a 20-year period, using a standardised chemistry setup (prescribed oxidants/photolysis rates) to isolate, predominantly, transport-driven variability between models. We examined the sensitivity of results to the choice of $CHBr_3/CH_2Br_2$ emission inventory within individual models, and also quantified the performance of emission inventories across the ensemble. The main findings of
TransCom-VSLS are summarised below.

– The TransCom-VSLS models reproduce the observed surface abundance, distribution and seasonal cycle of $CHBr_3$ and $CH_2Br_2$, at most locations where long-term measurements are available, reasonably well. At most sites, (i) the simulated seasonal cycle of these VSLS is not particularly sensitive to the choice of emission inventory, and (ii) the observed cycle is reproduced well simply from sea-
sonality in the chemical loss (a notable exception is at Mace Head, Ireland). Within a given model, absolute model-measurement agreement at the surface is highly dependent on the choice of VSLS emission inventory, particularly for $CHBr_3$ for which the global emission distribution and magnitude is somewhat poorly constrained. We find that at a number of locations, no consensus among models as to which emission inventory performs best can be reached. This is due to differences in the
representation/implementation of transport processes between models which can significantly influence the boundary layer abundance of short-lived tracers. This effect was observed between CTM variants which, other than tropospheric transport schemes, are identical. A major implication of this finding is that care must be taken when assessing the performance of emission inventories in order to constrain global VSLS emissions, based on single model studies alone. However, we also find
that within the tropics - where the troposphere-to-stratosphere transport of VSLS takes place - most models ($\sim$70%) achieve best agreement with measured surface $CHBr_3$ when using a bottom-up derived inventory, with the lowest $CHBr_3$ emission flux (Ziska et al., 2013). Similarly for $CH_2Br_2$ most

(also ~70%) of the models achieve optimal agreement using the $CH_2Br_2$ inventory with the lowest tropical emissions (Liang et al., 2010), though agreement is generally less sensitive to the choice of emission inventory (compared to $CHBr_3$). Recent studies have questioned the effectiveness of using aircraft observations and global-scale models (i.e. the top-down approach) in order to constrain regional VSLS emissions (Russo et al., 2015). For this reason and given growing interest as to possible climate-driven changes in VSLS emissions (e.g. Hughes et al., 2012), online calculations (e.g. Lennartz et al., 2015) which (i) consider interactions between the ocean/atmosphere state (based on observed seawater concentrations) and (ii) produce seasonally-resolved sea-to-air fluxes may prove a more insightful approach, over use of prescribed emission climatologies, in future modelling work.

– The TransCom-VSLS models generally agree on the locations where $CHBr_3$ and $CH_2Br_2$ are most elevated around the tropopause. These locations are consistent with known convectively active regions and include the Indian Ocean, the Maritime Continent and wider tropical West Pacific and the tropical Eastern Pacific, in agreement with of a number of previous VSLS-focused modelling studies (e.g. Aschmann et al., 2009; Pisso et al., 2010; Hossaini et al., 2012b; Liang et al., 2014). Owing to significant inter-model differences in transport processes, both the absolute tracer amount transported to the stratosphere and the amplitude of the seasonal cycle varies among models. However, of the above regions, the tropical West Pacific is the most important in all of the models (regardless of the emission inventory), due to rapid vertical ascent of VSLS simulated during boreal winter. In the free troposphere, the models reproduce observed $CHBr_3$ and $CH_2Br_2$ from the recent SHIVA and CAST campaigns in this region to within $\leq 16\%$ and $\leq 32\%$, respectively. However, at higher altitudes in the TTL the models generally (i) underestimated $CHBr_3$ between 14-16 km observed during the 2014 NASA ATTREX mission in this region but (ii) fell within $\pm 1$ $\sigma$ of the observed mean around the tropical tropopause (~17 km). Generally good agreement was also obtained to high altitude aircraft measurements of VSLS around the tropopause in the Eastern Pacific. During boreal summer, most models show elevated $CHBr_3$ around the tropopause above the Asian Monsoon region. However, the strength of this signal varies considerably among the models with a spread that encompasses virtually no $CHBr_3$ enhancement over the Monsoon region to strong (85%) $CHBr_3$ enhancements at the tropopause, with respect to the zonal average. Measurements of VSLS in the poorly sampled Monsoon region from the upcoming StratoClim campaign (http://www.stratoclim.org/) will prove useful in determining the importance of this region for the troposphere-to-stratosphere transport of VSLS.

– Climatologically, we estimate that $CHBr_3$ and $CH_2Br_2$ contribute 2.0 (1.2-2.5) ppt Br to the lower stratosphere through SGI, with the reported uncertainty due to the model spread. The TransCom-VSLS best estimate of 2.0 ppt Br is (i) ~57% larger than the measurement-derived best estimate of 1.28 ppt Br reported by Carpenter and Reimann (2014), and (ii) the TransCom-VSLS range (1.2-2.5 ppt Br) is ~37% smaller than the 0.6-2.65 ppt Br range reported by Carpenter and Reimann (2014). From this we suggest that, climatologically, the Carpenter and Reimann (2014) measurement-derived

SGI range, based on a limited number of aircraft observations (with a particular paucity in the tropical West Pacific), is potentially too conservative at the lower limit. Although we acknowledge that our uncertainty estimate (the model spread) does not account for a number of intrinsic uncertainties within global models, for example, tropospheric [OH] (as the models used the same set of prescribed oxidants). No significant transport-driven trend in stratospheric bromine SGI was found over the sim-

ulation period, though inter-annual variability was of the order of $\pm 5\%$. Loading of both $CHBr_3$ and $CH_2Br_2$ around the tropopause over the East Pacific is strongly coupled to ENSO activity but no strong correlation to ENSO or sea surface temperature was found when averaged across the wider tropical domain.

Overall, results from the TransCom-VSLS model intercomparison support the large body of ev-

765 idence that natural VSLS contribute significantly to stratospheric bromine. Given suggestions that VSLS emissions from the growing aquaculture sector will likely increase in the future (WMO, 2014; Phang et al., 2015) and that climate-driven changes to ocean emissions (Tegtmeier et al., 2015), tropospheric transport and/or oxidising capacity (Dessens et al., 2009; Hossaini et al., 2012a) could lead to an increase in the stratospheric loading of VSLS, it is paramount to constrain the present

770 day $Br_y^{VSLS}$ contribution to allow any possible future trends to be determined. In addition to SGI, this will require constraint on the stratospheric product gas injection of bromine which conceptually presents a number of challenges for global models given its inherent complexity.

*Acknowledgements.* RH thanks M. Chipperfield for comments and the Natural Environment Research Council (NERC) for funding through the TropHAL project (NE/J02449X/1). PKP was supported by JSPS/MEXT

KAKENHI-A (grant 22241008). GK, B-MS and PK acknowledge funding by the Deutsche Forschungsgemeinschaft (DFG) through the Research Unit SHARP (SI 1400/1-2 and PF 384/9-1 and in addition through grant PF 384/12-1) and by the Helmholtz Association through the Research Programme ATMO. NRPH and JAP acknowledge support of this work through the ERC ACCI project (project no. 267760), and by NERC through grant nos. NE/J006246/1 and NE/F1016012/1. NRPH was supported by a NERC Advanced Research Fellow-

ship (NE/G014655/1). PTG was also support through ERC ACCI. Contribution of JA and R Hommel has been funded in part by the DFG Research Unit 1095 SHARP, and by the German Ministry of Education and Research (BMBF) within the project ROMIC-ROSA (grant 01LG1212A).

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

**Table 1** Summary of the VSLS tracers simulated by the models, the global total emission flux (Gg VSLS $yr^{-1}$) and the rate constant for their reaction with OH (Sander et al., 2011). See text for details of emission inventories.

| Tracer # | Species | Tracer name | Ocean emission inventory | | Rate constant (VSLS + OH reaction) |
| | | | Global flux (Gg $yr^{-1}$) | Reference | k(T) (cm$^3$ molec$^{-1}$ s$^{-1}$) |
|---|---|---|---|---|---|
| 1 | Bromoform | CHBr$_3$_L | 450 | Liang et al. (2010) | $1.35\times10^{-12}$exp(-600/T) |
| 2 | | CHBr$_3$_O | 530 | Ordóñez et al. (2012) | |
| 3 | | CHBr$_3$_Z | 216 | Ziska et al. (2013) | |
| 4 | Dibromomethane | CH$_2$Br$_2$_L | 62 | Liang et al. (2010) | $2.00\times10^{-12}$exp(-840/T) |
| 5 | | CH$_2$Br$_2$_O | 67 | Ordóñez et al. (2012) | |
| 6 | | CH$_2$Br$_2$_Z | 87 | Ziska et al. (2013) | |

**Table 2** Overview of TransCom-VSLS models and model variants.

| # | Model[1] | Institution[2] | Resolution | | Meteorology[5] | BL mix. | Convection | Reference |
|---|---|---|---|---|---|---|---|---|
| | | | Horizontal[3] | Vertical[4] | | | | |
| 1 | ACTM | JAMSTEC | 2.8°×2.8° | 67$\sigma$ | JRA-25 | Mellor and Yamada (1974) | Arakawa and Shubert (1974) | Patra et al. (2009) |
| 2 | B3DCTM | UoB | 3.75°×2.5° | 40 $\sigma$-$\theta$ | ECMWF ERA-Interim | Simple[7] | ERA-Interim, archived[8] | Aschmann et al. (2014) |
| 3 | **EMAC**[6](_free) | KIT | 2.8°×2.8° | 39 $\sigma$-$p$ | Online, free-running | Jöckel et al. (2006) | Tiedtke (1989) [9] | Jöckel et al. (2006, 2010) |
| 4 | *EMAC* (_nudged) | KIT | 2.8°×2.8° | 39 $\sigma$-$p$ | Nudged to ERA-Interim | Jöckel et al. (2006) | Tiedtke (1989) [9] | Jöckel et al. (2006, 2010) |
| 5 | MOZART | EMU | 2.5°×1.9° | 56 $\sigma$-$p$ | MERRA | Holtslag and Boville (1993) | Note 10 | Emmons et al. (2010) |
| 6 | NIES-TM | NIES | 2.5°×2.5° | 32 $\sigma$-$\theta$ | JCDAS (JRA-25) | Belikov et al. (2013) | Tiedtke (1989) | Belikov et al. (2011, 2013) |
| 7 | STAG | AIST | 1.125°×1.125° | 60 $\sigma$-$p$ | ECMWF ERA-Interim | Taguchi et al. (2013) | Taguchi et al. (2013) | Taguchi (1996) |
| 8 | TOMCAT | UoL | 2.8°×2.8° | 60 $\sigma$-$p$ | ECMWF ERA-Interim | Holtslag and Boville (1993) | Tiedtke (1989) | Chipperfield (2009) |
| 9 | *TOMCAT* (_conv) | UoL | 2.8°×2.8° | 60 $\sigma$-$p$ | ECMWF ERA-Interim | Holtslag and Boville (1993) | ERA-Interim, archived[8] | Chipperfield (2009) |
| 10 | **UKCA** (_low) | UoC/NCAS | 3.75°×2.5° | 60 $\sigma$-z | Online, free-running | Lock et al. (2000) | Gregory and Rowntree (1990) | Morgenstern et al. (2009) |
| 11 | *UKCA* (_high) | UoC/NCAS | 1.875°×1.25° | 85 $\sigma$-z | Online, free-running | Lock et al. (2000) | Gregory and Rowntree (1990) | Morgenstern et al. (2009) |

[1] All models are offline CTMs except bold entries which are CCMs. Model variants are shown in italics.

CCMs ran using prescribed sea surface temperatures from observations.

[2] JAMSTEC: Japan Agency for Marine-Earth Science and Technology, Japan; UoB: University of Bremen, Germany; KIT: Karlsruhe Institute of Technology,

Germany; EMU: Emory University, USA ; NIES: National Institute for Environmental Studies, Japan; AIST: National Institute of Advanced

Industrial Science and Technology, Japan; UoL: University of Leeds, UK; UoC: University of Cambridge, UK; NCAS: National Centre for

Atmospheric Science, UK.

[3] Longitude×latitude

[4] $\sigma$: terrain-following sigma levels (pressure divided by surface pressure); $\sigma$-$p$: hybrid sigma-pressure; $\sigma$-$\theta$: hybrid sigma-potential

temperature; $\sigma$-z: hybrid sigma-height.

[5] MERRA: Modern-era Retrospective Analysis for Research and Applications; JCDAS: Japan Meteorological Agency Climate Data

Assimilation System; JRA-25: Japanese 25-year ReAnalysis; ECMWF: European Center for Medium range Weather Forecasting.

[6] ECHAM/MESSy Atmospheric Chemistry (EMAC) model (Roeckner et al., 2006). ECHAM5 version 5.3.02. MESSy version 2.42.

[7] Simple averaging of tracer mixing ratio below ERA-Interim boundary layer height.

[8] Read-in convective massfluxes from ECMWF ERA-Interim. See Aschmann et al. (2011) for B3DCTM implementation and Feng et al. (2011) for TOMCAT implementation.

[9] With modifications from Nordeng (1994).

[10] Shallow & mid-level convection (Hack , 1994); deep convection (Zhang and McFarlane, 1995).

**Table 3** Summary and location of ground-based surface VSLS measurements used in TransCom-VSLS, arranged from north to south. All sites are part of the NOAA/ESRL global monitoring network, with the exception of TAW, at which measurements were obtained by the University of Cambridge (see main text). *Stations SUM, MLO and SPO elevated at $\sim$3210m, 3397m and 2810m, respectively.

| Station | Site Name | Latitude | Longitude |
|---|---|---|---|
| ALT | Alert, NW Territories, Canada | 82.5° N | 62.3° W |
| SUM* | Summit, Greenland | 72.6° N | 38.4° W |
| BRW | Pt. Barrow, Alaska, USA | 71.3° N | 156.6° W |
| MHD | Mace Head, Ireland | 53.0° N | 10.0° W |
| LEF | Wisconsin, USA | 45.6° N | 90.2° W |
| HFM | Harvard Forest, USA | 42.5°N | 72.2° W |
| THD | Trinidad Head, USA | 41.0° N | 124.0° W |
| NWR | Niwot Ridge, Colorado, USA | 40.1° N | 105.6° W |
| KUM | Cape Kumukahi, Hawaii, USA | 19.5° N | 154.8° W |
| MLO* | Mauna Loa, Hawaii, USA | 19.5° N | 155.6° W |
| TAW | Tawau, Sabah, Malaysian Borneo | 4.2° N | 117.9° E |
| SMO | Cape Matatula, American Samoa | 14.3° S | 170.6° W |
| CGO | Cape Grim, Tasmania, Australia | 40.7° S | 144.8° E |
| PSA | Palmer Station, Antarctica | 64.6° S | 64.0° W |
| SPO* | South Pole | 90.0° S | - |

**Table 4** Correlation coefficient (r) between the observed and simulated climatological monthly mean surface CHBr$_3$ volume mixing ratio (at ground-based monitoring sites, Table 3). Model output based on CHBr$_3$_L tracer (i.e. using aseasonal emissions inventory of Liang et al. (2010)). Stations in bold denote where virtually all models fail to reproduce phase of the observed CHBr$_3$ seasonal cycle.

| Site | ACTM | B3DCTM | EMAC_F | EMAC_N | MOZART | NIES | STAG | TOMCAT | UKC_LO | UKCA_HI |
|------|------|--------|--------|--------|--------|------|------|--------|--------|---------|
| ALT | 0.91 | 0.90 | 0.89 | 0.89 | 0.95 | 0.93 | 0.60 | 0.94 | 0.92 | 0.94 |
| SUM | 0.69 | 0.73 | 0.71 | 0.70 | 0.84 | 0.71 | 0.40 | 0.73 | 0.75 | 0.88 |
| BRW | 0.96 | 0.97 | 0.89 | 0.91 | 0.99 | 0.98 | 0.73 | 0.97 | 0.94 | 0.97 |
| **MHD** | −0.89 | −0.89 | −0.93 | −0.89 | −0.85 | −0.89 | −0.79 | −0.90 | −0.91 | −0.73 |
| LEF | 0.84 | 0.72 | 0.74 | 0.78 | 0.83 | 0.74 | 0.35 | 0.43 | 0.78 | 0.88 |
| HFM | 0.64 | 0.61 | 0.66 | 0.69 | 0.79 | 0.46 | 0.08 | 0.58 | 0.40 | 0.81 |
| **THD** | −0.87 | −0.65 | −0.58 | −0.42 | 0.26 | −0.65 | −0.63 | −0.51 | −0.48 | −0.12 |
| NWR | 0.92 | 0.91 | 0.91 | 0.93 | 0.98 | 0.94 | 0.74 | 0.94 | 0.92 | 0.93 |
| KUM | 0.74 | 0.74 | 0.72 | 0.73 | 0.78 | 0.70 | 0.57 | 0.74 | 0.74 | 0.69 |
| MLO | 0.94 | 0.97 | 0.99 | 0.98 | 0.98 | 0.95 | 0.95 | 0.99 | 0.95 | 0.93 |
| TAW | −0.27 | −0.08 | 0.17 | −0.05 | −0.34 | −0.07 | −0.15 | 0.23 | 0.13 | 0.22 |
| SMO | 0.56 | 0.45 | 0.43 | 0.72 | 0.32 | 0.23 | 0.04 | 0.72 | 0.59 | −0.19 |
| CGO | −0.64 | 0.72 | −0.22 | −0.18 | −0.53 | 0.31 | 0.85 | −0.71 | −0.72 | −0.35 |
| PSA | 0.13 | 0.24 | 0.60 | 0.44 | 0.40 | -0.39 | 0.16 | 0.14 | 0.09 | 0.62 |
| SPO | 0.90 | 0.91 | 0.85 | 0.89 | 0.94 | 0.41 | 0.71 | 0.92 | 0.93 | 0.88 |

**Table 5** As Table 4 but for $CH_2Br_2$.

| Site | ACTM | B3DCTM | EMAC_F | EMAC_N | MOZART | NIES | STAG | TOMCAT | UKCA_LO | UKCA_HI |
|------|------|--------|--------|--------|--------|------|------|--------|---------|---------|
| ALT | 0.90 | 0.97 | 0.79 | 0.82 | 0.96 | 0.98 | 0.77 | 0.94 | 0.85 | 0.96 |
| SUM | 0.71 | 0.93 | 0.75 | 0.76 | 0.92 | 0.91 | 0.87 | 0.77 | 0.79 | 0.96 |
| BRW | 0.87 | 0.92 | 0.82 | 0.85 | 0.93 | 0.91 | 0.90 | 0.88 | 0.93 | 0.93 |
| **MHD** | −0.65 | −0.73 | −0.72 | −0.69 | −0.76 | −0.75 | −0.64 | −0.72 | −0.71 | −0.76 |
| LEF | 0.87 | 0.73 | 0.84 | 0.84 | 0.94 | 0.94 | 0.47 | 0.62 | 0.88 | 0.96 |
| HFM | 0.82 | 0.79 | 0.83 | 0.84 | 0.95 | 0.90 | −0.02 | 0.75 | 0.72 | 0.92 |
| THD | 0.54 | 0.80 | 0.73 | 0.79 | 0.78 | 0.84 | 0.04 | 0.69 | 0.66 | 0.75 |
| NWR | 0.90 | 0.88 | 0.91 | 0.89 | 0.99 | 0.97 | 0.86 | 0.91 | 0.92 | 0.97 |
| KUM | 0.90 | 0.89 | 0.90 | 0.91 | 0.99 | 0.91 | 0.74 | 0.90 | 0.92 | 0.98 |
| MLO | 0.90 | 0.89 | 0.94 | 0.91 | 0.96 | 0.90 | 0.30 | 0.91 | 0.93 | 0.97 |
| TAW | −0.83 | −0.80 | −0.78 | −0.75 | −0.39 | −0.47 | −0.12 | 0.15 | 0.20 | −0.16 |
| SMO | −0.08 | 0.67 | −0.14 | 0.59 | 0.38 | −0.12 | 0.34 | 0.97 | 0.74 | 0.00 |
| CGO | 0.59 | −0.43 | 0.45 | 0.30 | 0.64 | −0.06 | −0.42 | 0.80 | 0.80 | 0.41 |
| PSA | 0.17 | 0.71 | 0.52 | 0.68 | 0.75 | 0.08 | 0.62 | 0.72 | 0.65 | 0.68 |
| SPO | 0.88 | 0.91 | 0.82 | 0.86 | 0.95 | −0.04 | 0.97 | 0.90 | 0.94 | 0.88 |

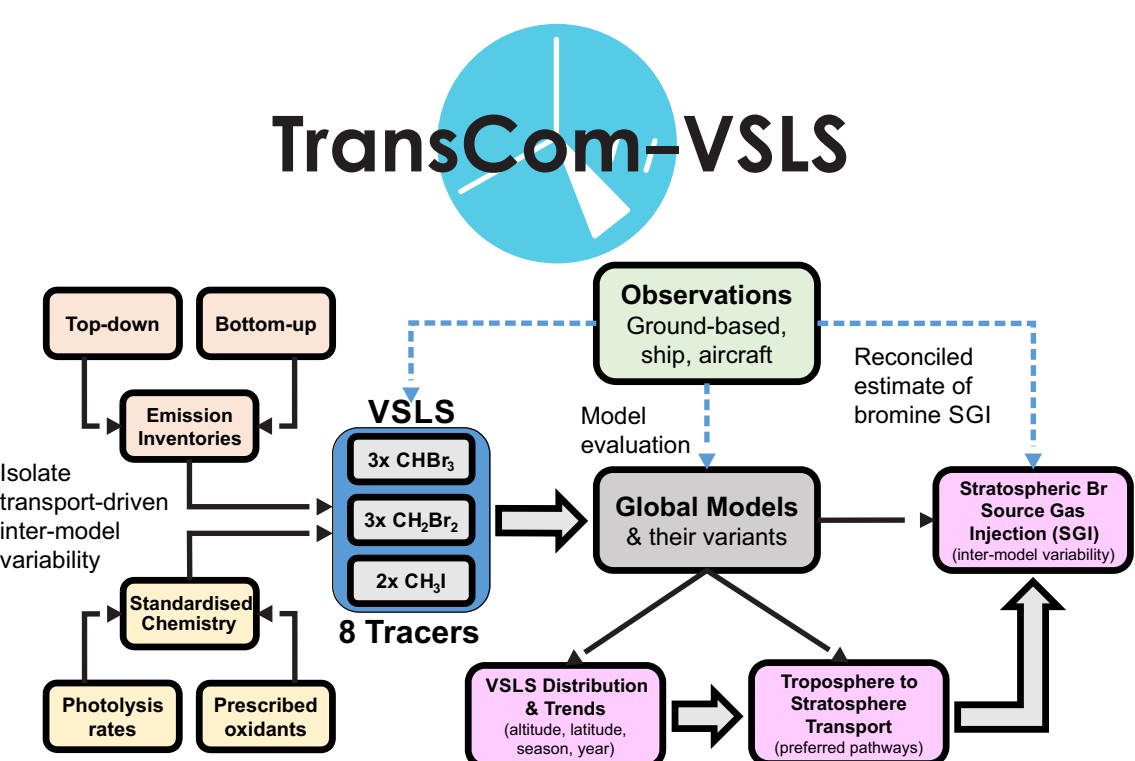

**Figure 1.** Schematic of the the TransCom-VSLS project approach.

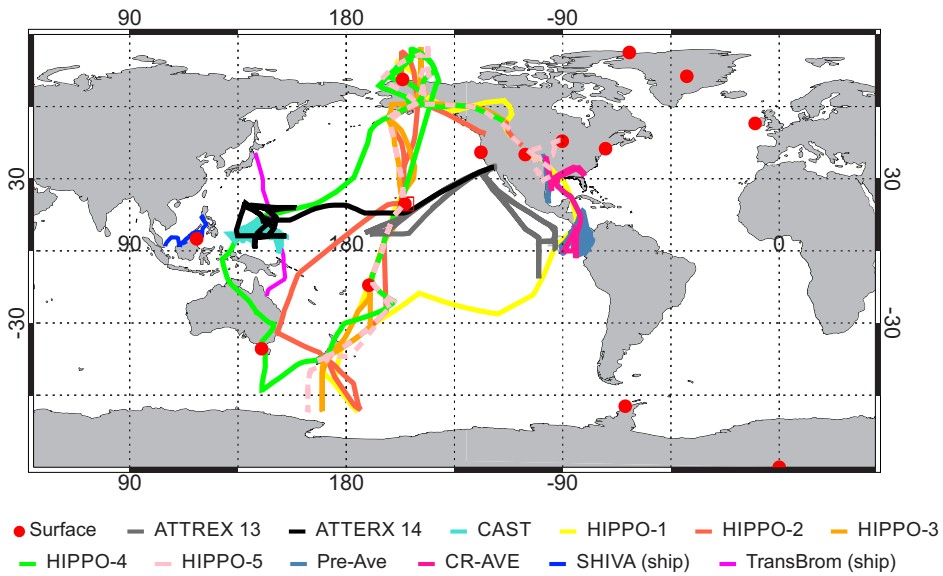

**Figure 2.** Summary of ground-based and campaign data used in TransCom-VSLS. See main text for details.

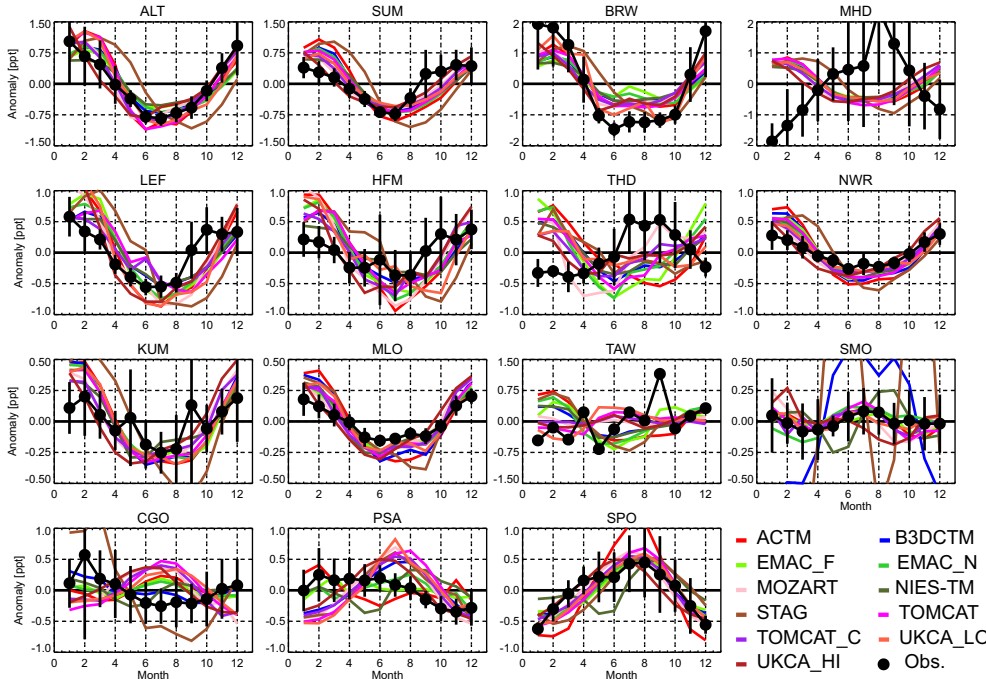

**Figure 3.** Comparison of the observed and simulated seasonal cycle of surface CHBr$_3$ at ground-based measurement sites (see Table 3). The seasonal cycle is shown here as climatological (1998-2011) monthly mean anomalies, calculated by subtracting the climatological monthly mean CHBr$_3$ mole fraction (ppt) from the climatological annual mean, in both the observed (black points) and model (coloured lines, see legend) data sets. The location of the surface sites is summarised in Table 3. Model output based on CHBr$_3$_L tracer (i.e. using aseasonal emissions inventory of Liang et al. (2010)). Horizontal bars denote $\pm 1\sigma$.

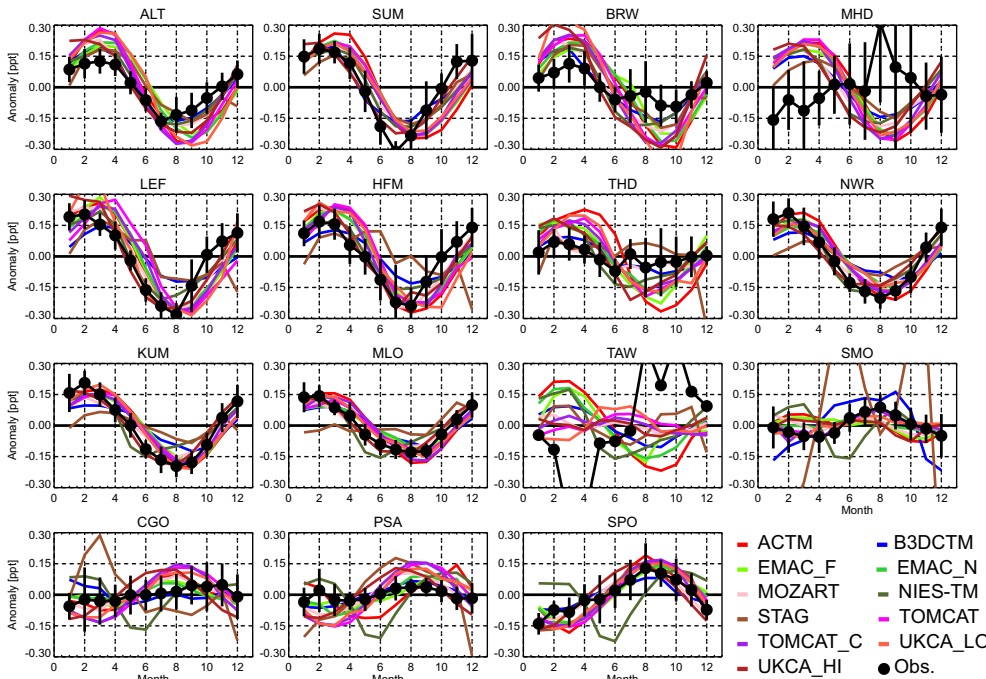

**Figure 4.** As Figure 3 but for $CH_2Br_2$.

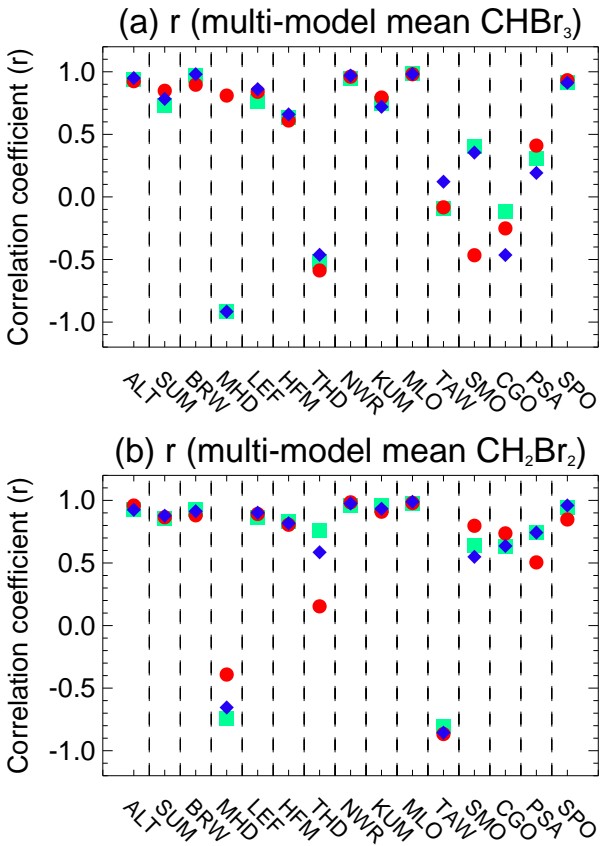

**Figure 5.** Correlation coefficient (r) between observed and multi-model mean (a) CHBr$_3$ and (b) CH$_2$Br$_2$, at ground-based monitoring sites. The correlation here represents the mean annual seasonal variation. At each site, 3×r values are given, reflecting the 3 different model CHBr$_3$ tracers; green squares denote the CHBr$_3$_L tracer (top-down derived Liang et al. (2010) emissions), blue diamonds denote the CHBr$_3$_O tracer (top-down Ordóñez et al. (2012) emissions) and red circles denote the CHBr$_3$_Z tracer (bottom-up Ziska et al. (2013) emissions).

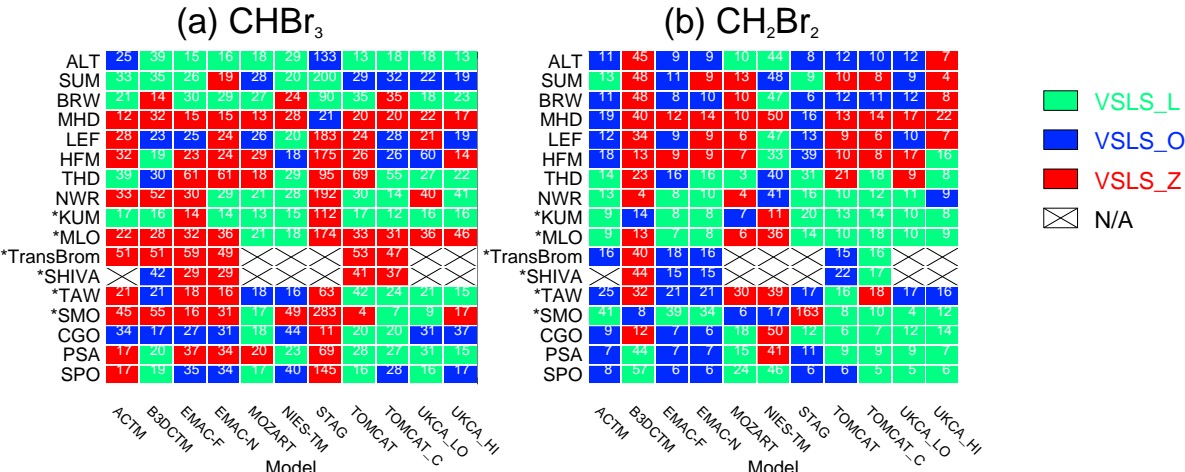

**Figure 6.** Summary of agreement between model (a) $CHBr_3$ and (b) $CH_2Br_2$ tracers and corresponding surface observations (ground-based, see Table 3, and TransBrom/SHIVA ship cruises). The fill colour of each cell (see legend) indicates the tracer giving the best agreement for that model, i.e. the lowest mean absolute percentage error (MAPE, see main text for details), and the numbers within the cells give the MAPE value (%), for each model compared to the observations. $CHBr_3$_L tracer used the Liang et al. (2010) emissions inventory, $CHBr_3$_O tracer used Ordóñez et al. (2012) and $CHBr_3$_Z tracer used Ziska et al. (2013). Sites marked with * are tropical locations. Certain model-measurement comparisons are not available (N/A).

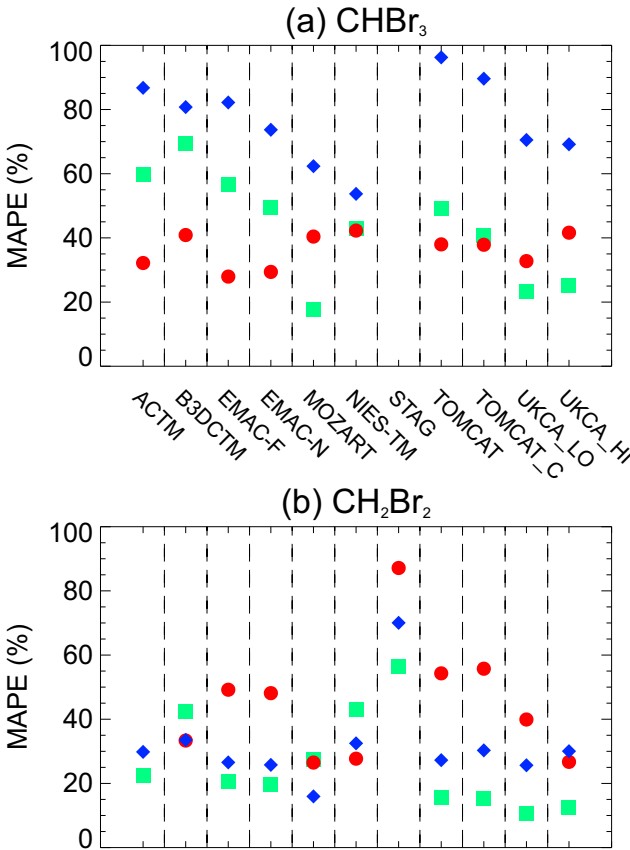

**Figure 7.** Overall mean absolute percentage error (MAPE) between model (a) $CHBr_3$ and (b) $CH_2Br_2$ tracers and corresponding surface observations, within the tropics only (i.e. sites KUM, MLO, TAW, SMO and the TransBrom and SHIVA ship cruises). Note, the scale is capped at 100%. A small number of data points fall outside of this range. Green squares denote the $CHBr_3\_L$ tracer, blue diamonds denote the $CHBr_3\_O$ tracer and red circles denote the $CHBr_3\_Z$ tracer.

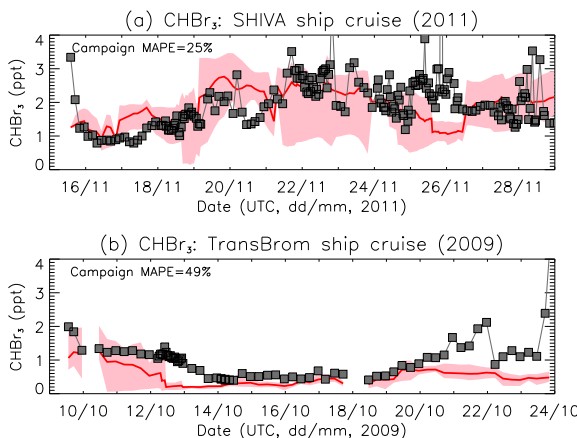

**Figure 8.** Comparison of modelled versus observed CHBr$_3$ surface volume mixing ratio (ppt) during (a) SHIVA (2011) and (b) TransBrom (2009) ship cruises. The multi-model mean is shown and the shaded region is the model spread. The mean absolute percentage error (MAPE) over each campaign is annotated.

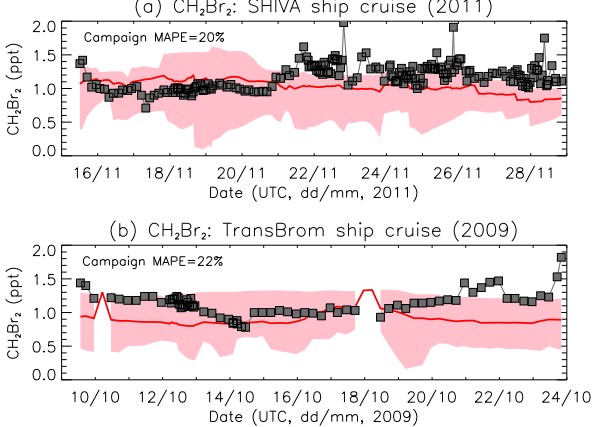

**Figure 9.** As Figure 8 but for CH$_2$Br$_2$.

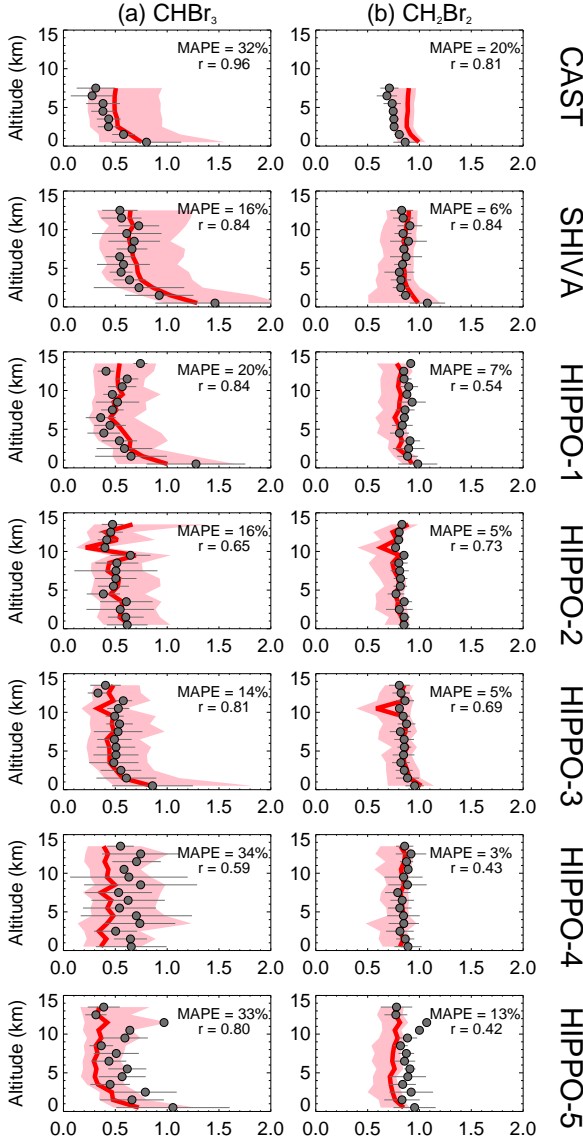

**Figure 10.** Compilation of modelled versus observed tropical profiles of (a) $CHBr_3$ and (b) $CH_2Br_2$ mixing ratio (ppt) from recent aircraft campaigns. Details of campaigns given in Section 2.4. Campaign mean observed profiles derived from tropical measurements only and averaged in 1 km vertical bins (filled circles). The horizontal bars denote $\pm 1\sigma$ from the observed mean. Shown is the corresponding multi-model mean profile (red) and model spread (shading). All models were included in the MMM with the exception of STAG (see Section 3.1.2). Models were sampled in the same space/time as the observed values, though for the comparison to CAST data, a climatological model profile is shown. The model-measurement correlation coefficient (r) and the mean absolute percentage error (MAPE, see main text) between the two are indicated in each panel.

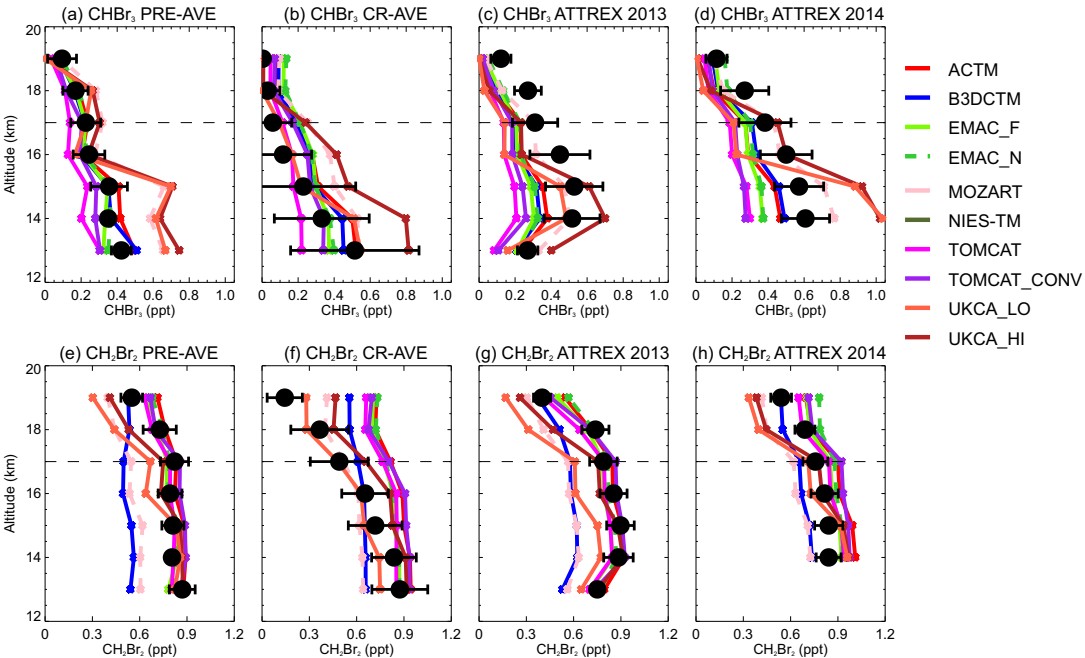

**Figure 11.** Comparison of modelled versus observed volume mixing ratio (ppt) of CHBr₃ (panels a-d) and CH₂Br₂ (panels e-h) from aircraft campaigns in the tropics (see main text for campaign details). The observed values (filled circles) are averages in 1 km altitude bins and the error bars denote $\pm 1\sigma$. The dashed line denotes the approximate cold point tropopause for reference.

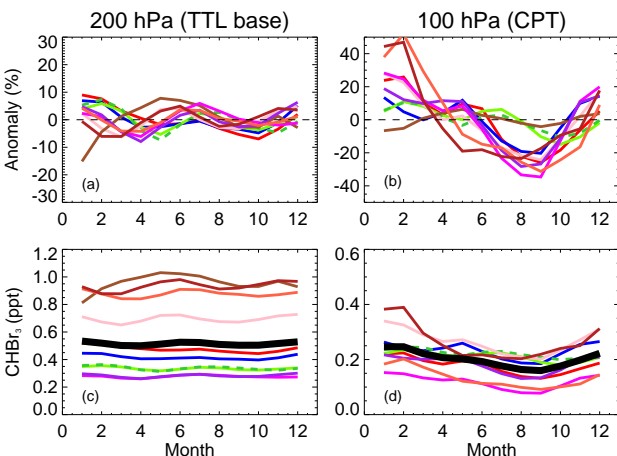

**Figure 12.** Simulated monthly mean anomalies of CHBr₃ volume mixing ratio (vmr), expressed as a percentage with respect to the annual mean, for (a) 200 hPa, the approximate base of the tropical tropopause layer (TTL) and (b) 100 hPa, the cold point tropopause (CPT). Panels (c) and (d) show the CHBr₃ vmr (ppt) at these levels. All panels show tropical ($\pm20°$ latitude) averages over the full simulation period (1993-2012). See Figure 3 for legend. Thick black line denotes multi-model mean.

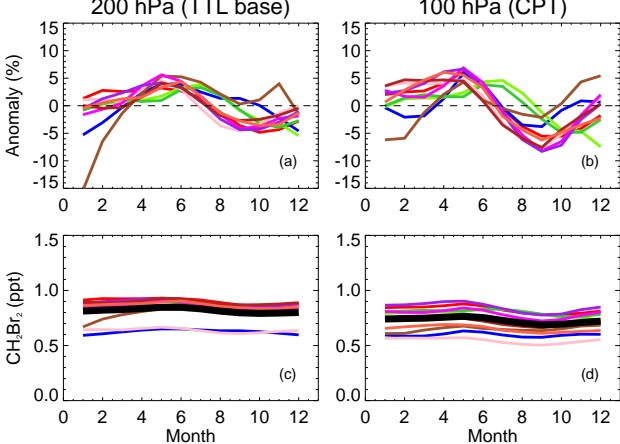

**Figure 13.** As Figure 12 but for CH₂Br₂.

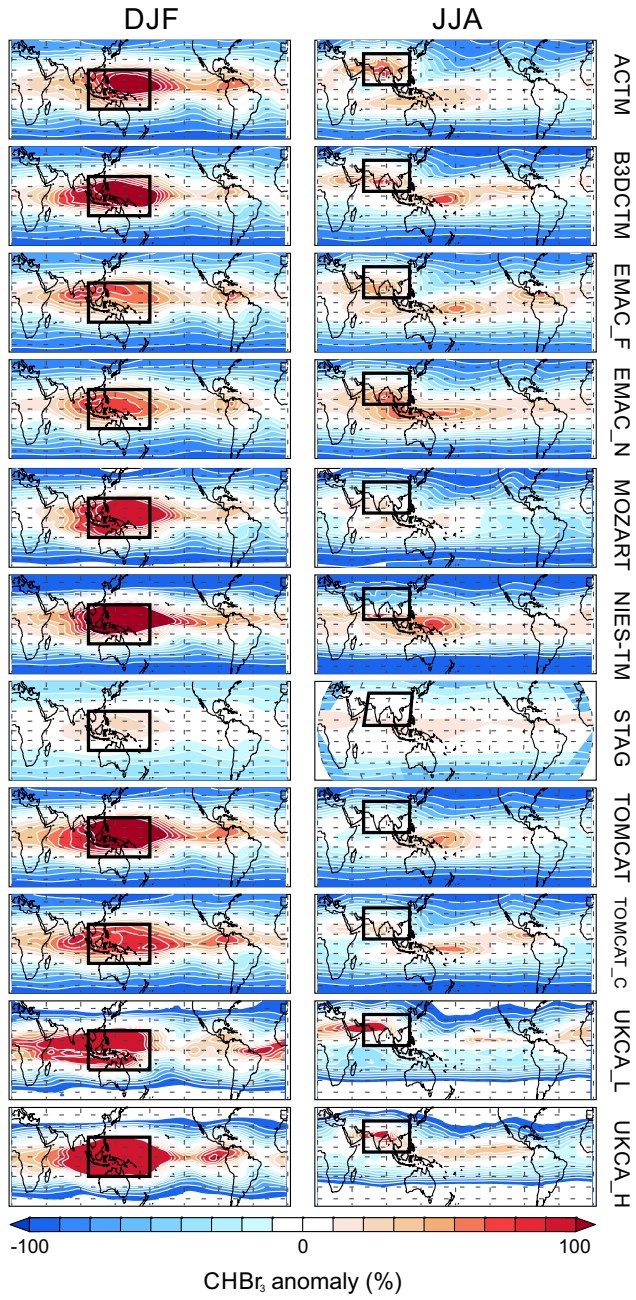

**Figure 14.** Simulated anomalies of the CHBr$_3$ volume mixing ratio with respect to the tropical ($\pm 30^\circ$ latitude) mean (expressed in %) at 100 hPa for (a) boreal winter (DJF) and (b) boreal summer (JJA). The boxes highlight the tropical West Pacific and location of the Asian Monsoon - regions experiencing strong convection.

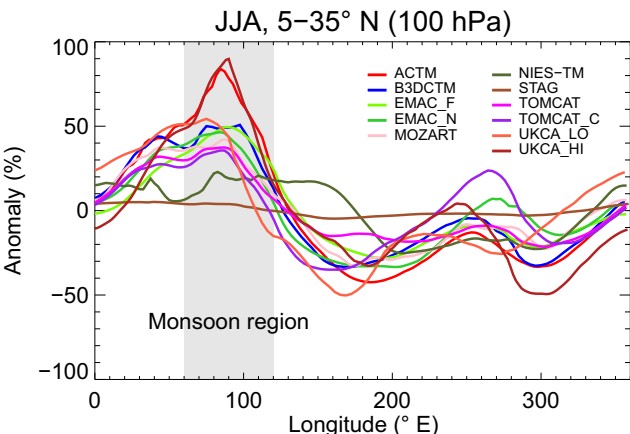

**Figure 15.** Simulated anomalies of the CHBr$_3$ volume mixing ratio at 100 hPa, as a function of longitude. Expressed as a percentage (%) departure from the mean within the latitude range of the Asian Monsoon (5°N-35°N), during boreal summer (JJA).

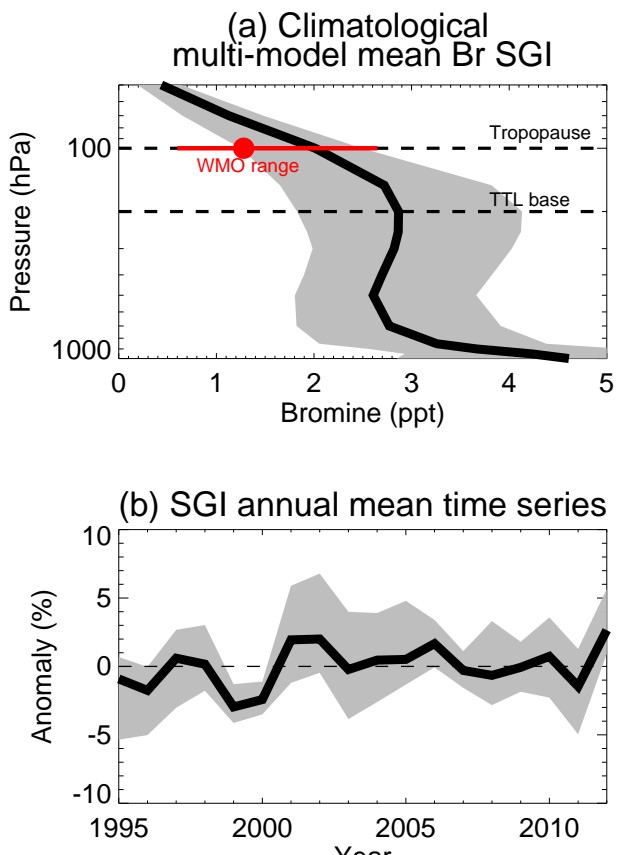

**Figure 16.** (a) climatological multi-model mean source gas injection of bromine (ppt) from $CHBr_3$ and $CH_2Br_2$ (i.e. $[3 \times CHBr_3] + [2 \times CH_2Br_2]$ mixing ratio). The shaded region denotes the model spread. Also shown is the best estimate (red circle) and SGI range from these gases (based on observations) reported in the most recent WMO $O_3$ Assessment Report (Carpenter and Reimann, 2014). (b) time series of multi-model mean stratospheric bromine SGI anomalies. Anomalies are calculated as the departue of the annual mean from the climatological mean (%).

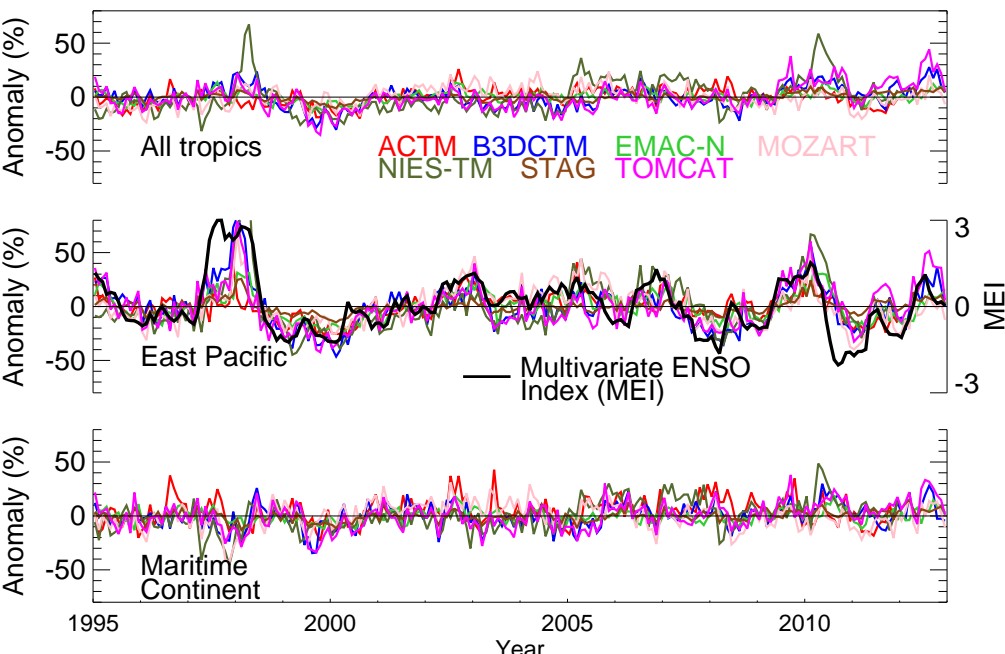

**Figure 17.** Monthly mean anomalies of CHBr$_3$ volume mixing ratio at 100 hPa, expressed as departures from the climatological monthly mean (%) over (a) tropical latitudes ($\pm 20°$), (b) the tropical East Pacific ($\pm 20°$ latitude, $180°$-$250°$E longitude) and (c) the Maritime Continent ($\pm 20°$ latitude, $100°$-$150°$E longitude). For the East Pacific region, the Multivariate ENSO Index (MEI) is also shown (see text). Note anomalies from free-running models not shown.