# Peer review of "A multi-model intercomparison of halogenated very short-lived substances (TransCom-VSLS): linking oceanic emissions and tropospheric transport for a reconciled estimate of the stratospheric source gas injection of bromine"

_Atmospheric Chemistry and Physics, 2015_

## Referee Comment (RC1) · Anonymous Referee #1 · 8 Feb 2016

I realize that this paper concerns a multi-model effort, and that it is difficult to analyze a large number of models and find common physical threads and results. Still I feel that this manuscript is lacking discussion of some things that raise obvious questions, and it would be worth the effort for the authors to make serious revisions.

Hossaini and co-authors describe a multi-model intercomparison that attempts to develop a reconciled estimate of the stratospheric injection of bromine. The paper is mainly descriptive. I suggest revisions that will place results in better context and

strengthen the paper.

Main suggestions for revision:

1) Rewrite the objectives. Although the paper meets the first two of the stated objectives (lines 95-100), the third and fourth objectives do not receive the attention of the first two. Objective (c) examines trends and inter-annual variability in the stratospheric loading of VSLS and (d) investigates how these relate to climate modes). The discussion of point (c) is limited to transport (mostly derived from assimilated meteorology) and point (d) is barely considered.

2) Include some discussion of CTM/CCM differences, and the factors that control whether or not CTMs with the same meteorological fields yield the same or similar results. Where different, the differences should be attributable to differences in CTM setup. Four of the 11 CTMs use ERA-Interim, and in addition, one version of EMAC is 'nudged' to ERM-interim. Two CTMS use JRA-25, one uses MERRA. There are three free running models; these will give similar results to the CTMs only if free running climatology is similar to the assimilated climatology. Although differences are said to be 'transport' – does that mean real differences in meteorology (e.g., differences between assimilation or free-running), differences in implementation of a single analysis, or differences among the analyses? When 70% of the models (or 8 of 11 models) do something does that mean the 8 models that use assimilation differ from the free running models? If convection and boundary layer mixing are dealt with differently among the CTMs, and are demonstrably different from the CCMs, then there should be some mention.

1) Include physical interpretation and a sense as to what we learn from 'lack of sensitivity of the simulated seasonal cycle to the choice of inventory' (line 615). If the mean value is sensitive to the inventory but the seasonal cycle is not, does that mean anything more compelling than that the seasonal cycle of the loss process (input to the simulations and the same in all models) is realistic?

**[ACPD](ACPD)**
2) Quantify the importance of SGI of VSLS to the total stratospheric bromine budget. It would be helpful to put the difference in SGI from WMO best estimate ($\sim$ 1.3 vs 2.0 (this work)) in the context of the stratospheric budget. IAV is +/- 5%? Is that significant? Is uncertainty in SGI more or less important than uncertainty in product gases? How large is the uncertainty in SGI + product gases relative to the total stratospheric bromine budget? Is the uncertainty in SGI + product gases smaller than the uncertainty in SGI?

3) How important is it that SGI does not show a transport trend? Isn't it just as likely that a trend (if any) would be due to a trend in the sources (as mentioned in penultimate paragraph)?

Specific Comments

Imprecise language throughout – paper has sufficient quantitative statements and comparisons that qualified descriptions detract from overall message. These are examples: 'reasonably well' 'not particularly sensitive' 'at most sites the amplitude of the seasonal cycle is generally consistent' 'to some degree likely reflects' 'likely' – followed by 5 references – how many do we need to make a concrete statement?

Why 'models are able to reproduce'? Why not 'models reproduce'?

You don't need to repeatedly say 'participating models' (unless you are also showing output from models that did not participate).

Abstract and Introduction Not clear until section 2.3 that most of the models are CTMs. Very surprising and possibly misleading that nothing is said about input meteorology. Differences among CTMs that use the same source for meteorology are differences in implementation since all of them would claim that they are trying to solve the same general equations with the same meteorological input.

Line 30 – transport driven variability in the annual mean SGI is 5% - why is that 'however'? Isn't that small?

Line 52 delete last phrase 'in recent years' – very long sentence already says 'recent'

[Figure]

Line 55 and following: why is it important to differentiate the product gas injection from source gas injection? Is the NET impact of VSLs (PGI and SGI) better constrained that SGI or PGI separately?

Line 64 – should be 'coincides'

Line 83 – it seems to me that the robust evaluation of the ACTUAL SGI needs observations, not just a concerted model evaluation. All the models could give the same answer (especially if they all use the same input meteorology) and data might reveal them all to be wrong.

Line 115 – you specify the chemistry – thereby ELIMINATING (rather the minimizing) its contribution to inter model differences. Also – since most of the models use ERA-interim, and there is no discussion of differences in its implementation, it is somewhat misleading to say that this study isolates differences due to transport processes.

Line 136 – is aseasonal the same as 'annual average'?

Section 2.1

It would be useful to have some visual comparison of the emissions (perhaps supplementary material)? The words don't give a sense of how large the differences in emissions are, and without that the sensitivity to emissions or lack thereof does not make sense.

Line 160 the words after the semi-colon should have a verb, or the sentence should be re-written without a semi-colon.

Line 184 'diagnosed convection' – do you mean used the standard parameterization for transport? Identified convection? Not clear.

Section 2.3 Did the CCMs use observed sea surface temperatures (relevant for El Nino)?

Section 3.1.1 Line 300 – MHD, THD, CGO, PSA - simulated seasonal cycles do not

agree with data – does the simulated seasonality look like the imposed seasonality of the loss terms? (If it does, then how can the models perform differently?)

Line 315 – Is it important that the observed annual variation is much smaller at SMO and CGO than at many of the other sites? Disregarding SMO (some really weird behavior), CGO and PSA amplitude greatly overestimated although shape is vaguely similar. Any commonalities among the 60% of models that do not correlate with observations > 0.5? Resolution? Meteorology? Transport scheme? Boundary Layer dynamics? Anything?

317 – virtually all do not reproduce – how about 'almost none of the models reproduce' or 'virtually all of the models fail

I don't understand the point of this discussion (lines 317 ff) At MHD, seasonality in the local emission flux is suggested to be the dominant factor controlling the seasonal cycle of surface CHBr3 (Carpenter et al., 2005). This leads to the observed summer maximum (as shown in Figure 3) and is not represented in the models' CHBr3_L tracer which, at the surface, is driven by the aseasonal emission inventory of Liang et al. (2010). Why did Carpenter et al. make that 'suggestion'? This sort of model can only do what you tell it, so if Carpenter et al. are correct – then you would never expect the models to do this. So then, what is the point of going to the MMM? Why aren't you discussing whether an emission inventory that has a seasonal element does better? It would make more sense if there was a better sense of the differences in inventories. Specifically – why would TWO aseasonal inventories give different answers at MHD, if the seasonality of the emissions is speculated to be a controlling factor?

Section 3.1.2 338 between a model value (M) and an observation (O), why parenthesis around 'for each model tracer'?

Figure 6 – ok these are the minimum percentages – but how does the reader know that the difference between a 'best' comparison and a comparison with one of the other inventories is significant?

[Figure]

350 I presume you don't get MAPE for both species with the same inventory because loss processes are different time scale? Replace low CHBr3 MAPE (good agreement), at a given location using a particular inventory does not necessarily mean a corresponding low CH2Br2 MAPE can be achieved using the same inventory, at that location. with At a given location low CHBr3 MAPE (good agreement) does not necessarily accompany a corresponding low CH2Br2 MAPE using the same inventory 355 – is this also related to how the inventories are created in the first place – i.e., how much do the inventories themselves depend on models and/or ERA-Interim? 365 – you attribute all differences to physical processes – e.g., convection and boundary layer mixing. Since most of these use assimilated meteorology, does that mean implementations differ among CTMs. Also, there are some pretty large differences between a free running simulation and a nudged simulation, so differences among 'variants' should not be surprising. Finally, in the prior modeling studies that had best agreement with different inventories, the loss terms were presumably different.

370 – why are differences in model variants surprising? In one case, this is the difference between free-running and nudged, and it is more likely than not that convection differs between these two in both intensity and location. In the second case, the chief difference that is discussed in convection, so again, performance is more likely to be different than it is to be the same.

395 It would be better to say 'For the N (fill in number) models that submitted hourly output . . . After that, paper says "Generally, the models reproduce the observed mixing ratios from SHIVA well, with a MMM campaign MAPE of 25% or less for both VSLS." This good agreement clearly depends on who is looking, and whether it makes sense to compute MAPE for the multimodel mean when the spread indicated by shading can be as much as 1 ppt (lowest value) and about 2 ppt (MMM) (top panel). It is also confusing since each model is using its 'preferred' inventory, and seriously in the real world there is only one actually set of emissions. In the best of circumstances, I think the MMM conceals physical differences and/or deficiencies in a subgroup of
models. Here, with each model using its 'preferred' inventory, I think it is nearly impossible to understand they significance of good or poor agreement with the MMM. Section 3.2 412 – by using the model 'preferred' inventory, what you are testing here is given surface values, how similar is the transport to higher levels to that inferred from observations in the real atmosphere. There are other ways to do this of course – in fact, looking at EACH tracer profile as a fraction of its near surface value might be even more instructive. Nonetheless – the discussion is convoluted and should be re-written to state the main (physical) point clearly. I presume 'paramerized transport schemes' later in this paragraph refers mainly to convection?

447 Only the number of flights controls the variability comparing Pre-AVE to CR-AVE? Nothing seasonal or spatial? Are the models sampled like the aircraft to produce average profiles? Is the error bar the range of values, the standard deviation? Would standard error of the mean be better? The correlation coefficient – is that the correlation for the whole profile? Isn't that guaranteed to be large since observed and simulated profiles general decrease with altitude?

ATTREX higher values at higher altitude 'possibly reflects the location'? Isn't this true (and backed by other observations?) If it is only 'possible', what are the other causes? grammar - CR-AVE had nearly twice the number of flights AS Pre-Ave and . . .

Section 3.3

470 ' likely reflects the location at which the measurements were made' Why so many words, why 'likely' (what else could it be) and why no direct statement about zonal asymmetry? Would the model zonal means compare better with Carpenter and Reimann? Or should it be model mean in a different region compared with Carpenter and Reimann?

Section 3.4

515 If most of the models are using assimilated fields, how can they fail to locate

the areas of deep convection and the seasonal dependence therein? It is all right to describe this behavior, but I would hardly call it a prediction. Too much discussion, especially since the result is not novel.

525 – variations in the importance of Monsoon – any connection to the input data or model type? I don't think this is evidence that UKCA-HI has a more faithful representation of convection – you would need some other information about HI vs LOW to make this statement.

Section 4

605 previously when you talked about variability it was physical (e.g., seasonal etc. – something real). Here you are talking about differences among models for different inventories. It is confusing to call this 'variability'.

622 – model variants are identical except for tropospheric transport schemes. Based on everything else written, I don't think this statement is correct. E.g, the tropospheric transport of 'nudged' vs 'free-running' will differ for physical reasons, not just 'transport schemes'.

The problem with single model studies of inventories or deriving inventories is that they don't typically include model error. If they did then the inventories would be more robust – or the differences among studies would likely fall within the errors.

625 – For both CHBr3 and CH2Br2 the 'best' inventory for the tropics is the lowest – but at the same time agreement here is 'less sensitive to choice of inventory'. What point are you trying to make? I don't see how the statements about seasonally resolved air-to-sea fluxes follow from anything in this paragraph (noting this is the 'discussion and conclusions' section).

665 – the very long sentence beginning 'Although . . . ' should be clarified.

Picky comment Example: Do you really need to include so many references – e.g., five references to say that Bromine + chlorine destroys ozone more than chlorine by itself?

There are other examples of many references fo

---

## Referee Comment (RC2) · Anonymous Referee #2 · 14 Feb 2016

**General**

This paper presents a comprehensive model intercomparison of the impact of bromine containing VSLS on the stratospheric bromine loading. This is a good initiative and the outcome of this intercomparison will be important of assessing the impact of bromine on stratospheric ozone. It is in particular noteworthy that a lot of observations are employed to assess the quality of the model results.

I have some points (see below), where I think the discussion in the paper can be clarified and improved. The impact of particular model features (e.g., the convective schemes employed) on the different results could be brought out more clearly. The reader ultimately will be interested in what the problematic model features are, because these are the features that need improvement in the further developments of such models. This point cold be brought across in the paper in a better way.

In summary, I think that a revised version of the paper, taking into account the points raised in the reviews will be a valuable contribution to ACP.

**Detailed comments**

Five out of the 11 participating models are nudged to or driven by ERA-Interim. While ERA-Interim is a good choice, this fact will lead to the multi-model mean being biased to an ERA-Interim world. I suggest to bring this point across more clearly. Does this fact have any implications for the conclusions of this model intercomparison?

Another model feature, which is important for tropospheric transport of VSLS is the convective parametrisation used in the model (see for example Rybka, H. and Tost, H.: Uncertainties in future climate predictions due to convection parameterisations, Atmos. Chem. Phys., 14, 5561-5576, doi:10.5194/acp-14-5561-2014, 2014, and references therein). I suggest more discussion of this point in the paper. Also, the information of the convective scheme used in the different models should be included in Table 2. Perhaps some of the model differences and some of the model similarities can be attributed to using a particular convective parametrisation or a particular meteorology?

I also have reservations about the concept of a "preferred" tracer. I think this means that the emission inventory somehow interacts with the transport scheme of the model to produce reasonable results at higher altitudes. But this means that the higher altitude agreement could be right for the wrong reason. I know it is demanding a lot from
models, but of course one would expect to design independently the best emission inventory and the best (vertical) transport to obtain the best agreement with measurements. Obviously this model intercomparison cannot achieve this goal, but I think the discussion of these issues could be improved.

Finally, the impact of ENSO activity on the stratospheric bromine loading is unclear. What is the message of the paper here? The paper states that there is a strong correlation of SGI with ENSO (e.g. abstract), but that there is no correlation of ENSO (MEI) with the bromine loading in the LS (e.g. conclusions). But SGI is important for the bromine loading in the LS. This points needs to be clarified and better discussed in the paper.

**Minor issues**

- Title: I am not sure if "TransCom-VSLS" should be in the title; the name of the project will not be relevant on a timescale of years, when the paper will still be read.

- l. 7: I do not think that model estimates should be used to "constrain" measurements.

- line 20: change 'optimal' to 'best'

- l. 36: Isn't 6 month a bit long for very short lived?

- l 51: 'recent' twice in this sentence

- l. 52: try \mathrm{VSLS} to avoid italics in VSLS. (Similar for MAPE (l. 345) below).

- l. 59: 'owing to' instead of 'due to'

- l. 76: I think you mean Tissier and Legras here

- l. 78: do you mean "broadly similar" here?

- l. 100: what do you mean by "climate modes" – more explanation here.

- Figure 1: This figure is not really discussed in the paper. Which message does it communicate? I suggest removing the figure from the paper.

- l. 144: is a bottom-up . . .

- line 179: this means that the multi-model mean is highly influenced by CTMs driven by ERA-Interim data – correct?

- l. 211: instead of 'see also' you could perhaps state for which information which paper should be consulted.

- l 301: what is the reason that 'clear outliers' are found? Are these models with obvious errors?

- l. 329: use $r$ for the correlation coefficient

- l. 366: why does convection influence "near-surface" abundances of VSLS?

- l. 414: I think it is problematic that models have a *preferred tracer*: doesn't this imply that results could be right for the wrong reason?

- l. 425: Where is the reproduction of the c-shape shown? This seems an important issue.

- l. 435: The concept of a 'preferred' tracer means that the emission inventory somehow interacts with the models transport scheme to produce reasonable results at higher altitudes – correct? Can you describe in more detail here, what 'worse agreement' means?

- l. 485: is CO really short-lived?

- l. 492: state the lifetime in months/weeks

- l. 527: you might want to add here also Tissier and Legras 2015; Vogel et al. 2014

- l 560: Clarify which best estimate is meant here, TransCom or WMO.

- l. 593-595: The last sentence states that the VSLS loading in the LS is not correlated to MEI. But the sentences above state that bromine SGI *is* sensitive to modes such as MEI. Isn't this a contradiction? I think more discussion is require here.

- l. 598: change to: these processes

- l. 599: change 'a range' to 'a number'

- l. 614-618: Is the point here that the seasonal cycle is not dependent on the emission inventory, but the absolute model-measurement agreement is? How can this be the case. Please clarify. (See also abstract).

- l. 626: change optimal to best

- l. 634: what exactly is meant by 'online calculations'?

- l. 648: But the 'higher altitudes' are most relevant for the transport of VSLS into the stratosphere – correct?

- l. 663: You mean the SGI range by Carpenter and Reiman, add the citation for clarification.

- l. 670-672: This is astonishing, isn't it? I suggest somewhat more discussion on this point.

- l. 676: change 'changes to' to 'changes of'

- l. 678: change 'increased' to 'increase of the'

- l. 679: distinguished from what?

- l. 689: why is R Hommel not abbreviated?

- Fig. 1: not sure if this figure is necessary

- Fig. 2: Continents in light grey would look better than in black.

- References: There are some references that need to be updated; ACP vs ACPD, Werner et al., 2016 etc.

---

## Author Comment (AC1) · 3 May 2016

Our responses to review comments (repeated in *italics*) are given below in **red**.

**Response to Reviewer 1**

*I realize that this paper concerns a multi-model effort, and that it is difficult to analyze a large number of models and find common physical threads and results. Still I feel that this manuscript is lacking discussion of some things that raise obvious questions, and it would be worth the effort for the authors to make serious revisions.*

*Hossaini and co-authors describe a multi-model intercomparison that attempts to develop a reconciled estimate of the stratospheric injection of bromine. The paper is mainly descriptive. I suggest revisions that will place results in better context and strengthen the paper.*

We thank the Reviewer for their comments on our manuscript. Many of the comments raised by the Reviewer involve clarification of points and further discussion. We have addressed these suggestions and believe the paper has been strengthened accordingly.

*Main suggestions for revision:*

*1) Rewrite the objectives. Although the paper meets the first two of the stated objectives (lines 95-100), the third and fourth objectives do not receive the attention of the first two. Objective (c) examines trends and inter-annual variability in the stratospheric loading of VSLS and (d) investigates how these relate to climate modes). The discussion of point (c) is limited to transport (mostly derived from assimilated meteorology) and point (d) is barely considered*

OK, we have reworked the objectives as the Reviewer requests. The objectives now better reflect the focus of this work.

*2) Include some discussion of CTM/CCM differences, and the factors that control whether or not CTMs with the same meteorological fields yield the same or similar results. Where different, the differences should be attributable to differences in CTM setup. Four of the 11 CTMs use ERA-Interim, and in addition, one version of EMAC is 'nudged' to ERM-interim. Two CTMS use JRA-25, one uses MERRA. There are three free running models; these will give similar results to the CTMs only if free running climatology is similar to the assimilated climatology.*

OK, we have added text throughout the manuscript to address these comments. Regarding model differences, we have added the convection and boundary layer parameterisation used by each model to Table 2 (as was also requested by Reviewer 2).

Specifically, regarding whether or not CTMs with the same meteorological fields yield similar results, we have added text to Section 3.1.1 discussing the cause of "outliers". We have also added text to Section 3.1.2 comparing, as an example, TOMCAT vs. B3DCTM at the surface (both models use ECMWF ERA-Interim). Note, it is not surprising that CTMs which use the same input meteorological fields can look different in terms of the simulated distribution of VSLS (or other species, as observed in previous TransCom experiments). This is because of differences in each model's parameterisation of convection and boundary layer mixing (which use the input meteorological fields in different ways and with different assumptions). As noted, details of these parameterisation are now given in Table 2. We already point to various papers in Section 3.1.2 that have shown large differences in the simulated near surface abundance of short-lived tracers due to difference in the treatment of the above transport processes.

In the revised manuscript, we have also expanded upon the discussion of convection as a large contributor to the simulated levels of CHBr$_3$ around the tropopause. For example, in Section 3.4 in discussion of Figures 14 and 15, we have added the following paragraph.

"The high altitude model-model differences in CHBr$_3$, highlighted in Figures 14 and 15, are attributed predominately to differences in the treatment of convection. Previous studies have shown that (i) convective updraft mass fluxes, including the vertical extent of deep convection

(relevant for bromine SGI from VSLS), vary significantly depending on the implementation of convection in a given model (e.g. Feng et al., 2011) and (ii) that significantly different short-lived tracer distributions are predicted from different models using different convective parameterisations (e.g. Hoyle et al., 2011). Such parameterisations are often complex, relying on assumptions regarding detrainment levels, trigger thresholds for shallow, mid-level and/or deep convection, and vary in their approach to computing updraft (and downdraft) mass fluxes. Furthermore, the vertical transport of model tracers is also sensitive to interactions of the convective parameterisation with the boundary layer mixing scheme (also parameterised) (Rybka and Tost, 2014). On the above basis and considering that the TransCom-VSLS models implement these processes in different ways (Table 2), it was not possible to detangle transport effects within the scope of this project. However, no systematic similarities/differences between models according to input meteorology were apparent".

The Reviewer asks that differences between models are attributed to "CTM setup". The paragraph beginning "As the chemical sink of VSLS…" in Section 3.1.2 has been appropriately reworked to reflect this comment (see response below). The text has also been amended in the Abstract and Conclusions so that it is clear that "implementation" of transport is important.

*Although differences are said to be 'transport' – does that mean real differences in meteorology (e.g., differences between assimilation or free-running), differences in implementation of a single analysis, or differences among the analyses? When 70% of the models (or 8 of 11 models) do something does that mean the 8 models that use assimilation differ from the free running models?*

We believe this is a terminology issue that simply needs clarification. As it was impossible here to isolate the exact cause of the differences between models, nor is that the focus of this work, the word "transport" is used to incorporate all the factors that the Reviewer states (considering that all models used common fluxes & chemistry). To clarify this and address the Reviewer's comment we have:

1. Defined what we mean by "transport" early in the manuscript: "…the effects of PBL mixing, convection and advection, and the implementation of these processes" – see introduction, final paragraph.

2. Noted in the Abstract and Conclusions that the implementation of transport processes is a clear factor (as each model would say that they are simulating convection, MBL mixing, advection etc.).

3. Reworked the appropriate paragraphs in Sections 3.1.2 and 3.4, where transport is discussed, to make clear that the "implementation" of transport and "CTM setup" are contributing factors. Throughout the manuscript more discussion is now given on the reasons for model differences.

*If convection and boundary layer mixing are dealt with differently among the CTMs, and are demonstrably different from the CCMs, then there should be some mention.*

As noted, we now include the convection and boundary layer mixing scheme used by each model in Table 2.

*1) Include physical interpretation and a sense as to what we learn from 'lack of sensitivity of the simulated seasonal cycle to the choice of inventory' (line 615). If the mean value is sensitive to the inventory but the seasonal cycle is not, does that mean anything more compelling than that the seasonal cycle of the loss process (input to the simulations and the same in all models) is realistic?*

That is correct, we are essentially saying here that the seasonal cycle of the loss process is accurate based on the various model-measurement comparisons at the surface. There is already some discussion on this point in Section 3.1.1. However, we have now added a sentence to make the point in the 2nd paragraph of Summary and Conclusions also, where the reviewer suggests.

*2) Quantify the importance of SGI of VSLS to the total stratospheric bromine budget. It would be helpful to put the difference in SGI from WMO best estimate (~1.3 vs 2.0 (this work)) in the context of the stratospheric budget. IAV is +/- 5%? Is that significant?*

We have addressed these comments and added some new discussion in Section 3.5 of the revised manuscript.

The reviewer comment refers to the TransCom multi model mean SGI of bromine from $CHBr_3$ and $CH_2Br_2$ (2.0 ppt Br) versus WMO best estimate of the same quantity derived from observations (1.28 ppt Br). In the context of the total stratospheric $Br_y$ budget, estimated to be ~20 ppt Br in 2011 (WMO, 2014), SGI of $CHBr_3$ and $CH_2Br_2$ accounts for 10% of this total (our estimate) versus 6.4% based on the current WMO best estimate.

In the context of total stratospheric $Br_y$ (~20 ppt), the ±5% IAV of modelled bromine SGI from $CHBr_3$ and $CH_2Br_2$ is small (sub ppt). We have made this point in the revised manuscript.

*Is uncertainty in SGI more or less important than uncertainty in product gases? How large is the uncertainty in SGI + product gases relative to the total stratospheric bromine budget?*

The uncertainty in SGI and PGI are similar. The WMO quote an uncertainty range of 0.7-3.4 ppt Br for SGI and 1.1-4.3 ppt Br for PGI (note, from all brominated VSLS). Their best estimate for SGI + PGI is therefore 2-8 ppt Br, giving an uncertainty of 6 ppt Br. Of this uncertainty PGI contributes 54% and SGI 46%.

The uncertainty in SGI + PGI (i.e. 6 ppt Br, above) corresponds to ~30% of the total stratospheric bromine budget (i.e. 100*[6 ppt / 20 ppt]).

*Is the uncertainty in SGI + product gases smaller than the uncertainty in SGI?*

No. The WMO uncertainty of SGI + PGI is 6 ppt Br. The WMO uncertainty in SGI is 2.7 ppt Br (i.e. 3.4 minus 0.7 ppt) and is therefore smaller.

We note additionally that we have amended the text in Section 3.5 to make clearer the reduction in SGI range the TransCom results suggest. In summary:

WMO SGI range ($CHBr_3$ + $CH_2Br_2$ only): 0.6 to 2.65 ppt Br

WMO SGI range (minor VSLS** only): 0.08 to 0.71 ppt Br

WMO SGI range (total, all VSLS): 0.7 to 3.4 ppt Br

TransCom considered $CHBr_3$ and $CH_2Br_2$ only, which dominate the SGI uncertainty range (based on the above numbers).

TransCom SGI range ($CHBr_2$ + $CH_2Br_2$ only): 2.0 (1.2 to 2.5) ppt Br

Therefore, if we take the WMO SGI range for the "minor" VSLS, the total TransCom SGI range (all VSLS) would be 1.28 to 3.21 ppt Br. This range is 28% smaller than the total WMO SGI range. This result is now incorporated into the text.

**including: $CHBr_2Cl$, $CH_2BrCl$, $CHBrCl_2$, $C_2H_5Br$, $C_2H_4Br$, and $C_3H_7Br$

*3) How important is it that SGI does not show a transport trend? Isn't it just as likely that a trend (if any) would be due to a trend in the sources (as mentioned in penultimate paragraph)?*

There are various factors that could, in principle, cause a trend in stratospheric bromine SGI. These could potentially include a trend in sources (e.g. oceanic emissions), as the reviewer notes, or a trend in the atmospheric loss rate of VSLS (e.g. due to any oxidant changes), or a trend in transport processes (e.g. convection). To our best knowledge, there is no strong evidence that any of these factors has caused a trend in the SGI of $CHBr_3$ and $CH_2Br_2$ over the study period.

One aim of this work was to isolate any possible transport trend impacting SGI. Therefore, VSLS emissions and the chemical loss of VSLS was fixed in each year, with no inter-annual variability. The fact no SGI trend was found over the study period does not preclude future SGI changes driven by climate-driven transport changes (or other factors). We already make this point in the final paragraph of the paper but have now also included it briefly in Section 3.5.

***Specific Comments***

*Imprecise language throughout – paper has sufficient quantitative statements and comparisons that qualified descriptions detract from overall message. These are examples: 'reasonably well' 'not particularly sensitive' 'at most sites the amplitude of the seasonal cycle is generally consistent' 'to some degree likely reflects' 'likely' – followed by 5 references – how many do we need to make a concrete statement?*

OK. We have been through the manuscript and addressed this by removing unnecessary qualified descriptions.

*Why 'models are able to reproduce'? Why not 'models reproduce'?*

OK. We have changed this throughout the manuscript.

*You don't need to repeatedly say 'participating models' (unless you are also showing output from models that did not participate).*

OK. We removed 'participating' in virtually all instances.

*Abstract and Introduction Not clear until section 2.3 that most of the models are CTMs. Very surprising and possibly misleading that nothing is said about input meteorology. Differences among CTMs that use the same source for meteorology are differences in implementation since all of them would claim that they are trying to solve the same general equations with the same meteorological input.*

OK. We now note in the abstract that most models are CTMs. We agree that differences in CTM implementation should be commented on and have also added to the abstract: "Overall, our results do not show systematic differences between models specific to the choice of reanalysis meteorology, rather clear differences are seen related to differences in the implementation of transport processes in the models". See earlier responses to comments also.

*Line 30 – transport driven variability in the annual mean SGI is 5% - why is that 'however'? Isn't that small?*

OK, we have removed 'however'. (See above IAV discussion).

*Line 52 delete last phrase 'in recent years' – very long sentence already says 'recent'*

OK, we have deleted that phrase.

*Line 55 and following: why is it important to differentiate the product gas injection from source gas injection? Is the NET impact of VSLs (PGI and SGI) better constrained that SGI or PGI separately?*

It is common to differentiate SGI from PGI in the literature concerning VSLS and it has indeed been done in all recent WMO Ozone reports. It is important here because our paper deals with SGI only. More broadly, this distinction is sensible given the differences in the two processes and how they can be quantified. For example, it is easier to quantify bromine SGI, from measurements of $CHBr_3$ and $CH_2Br_2$ etc. in the UT, than it is to quantify PGI, which requires measurements of $Br_y$ species (e.g. BrO, HBr). Observations alone cannot determine whether $Br_y$ has come from VSLS or other source gases. In addition, it is useful to consider SGI separately, as modelling work has shown that the stratospheric SGI of bromine from VSLS may increase in response to climate change (Hossaini et al. 2012a). To our knowledge, this has yet to be shown for PGI.

Regarding the SGI/PGI uncertainty relative to the total (SGI + PGI), we have already answered this point (see above responses).

*Line 64 – should be 'coincides'*

OK, this has been corrected.

*Line 83 – it seems to me that the robust evaluation of the ACTUAL SGI needs observations, not just a concerted model evaluation. All the models could give the same answer (especially if they all use the same input meteorology) and data might reveal them all to be wrong.*

As VSLS observations are limited in their space/time coverage, clearly the best approach is to consider both observations and models when evaluating SGI. That is exactly what we have done in this paper, therefore we are somewhat puzzled by this review comment. A wealth of observational data are used to evaluate the models and the model SGI results are compared to the existing measurement-derived SGI estimates from WMO.

Use of models to make predictions regarding future climate-driven changes to the stratospheric SGI of bromine from VSLS is increasing (e.g. Dessler et al. 2009; Hossaini et al. 2012a). Therefore, it is important that studies such as TransCom evaluate how these models perform in the present day. It has already been discussed above that input meteorology is not all that matters for tracer/VSLS transport in CTMs.

*Line 115 – you specify the chemistry – thereby ELIMINATING (rather the minimizing) its contribution to inter model differences. Also – since most of the models use ERAinterim, and there is no discussion of differences in its implementation, it is somewhat misleading to say that this study isolates differences due to transport processes.*

We use the work "minimising" here as it cannot be said that the chemical loss rates are *totally* identical as VSLS oxidation by OH is temperature dependent. Models use different reanalysis products (5 out of the 11 CTMs/nudged CCMs use ECMWF Era-Interim) and are expected to have similar, but not identical, temperature fields. Some differences in the chemical loss rate are therefore unavoidable, even when oxidants are prescribed in each model.

As can be seen from Figure 14, the model-model differences are quite large due to transport (mainly; only a second order effect in loss due temperature). Even the models using ERA-interim (B3DCTM, EMAC-nudge, TOMCATs) show large differences at 100 hPa. Thus the ensemble of 11 models do account for a large fraction, if not most, of the uncertainty due to model transport or implementation of such. As already noted, differences in the treatment of convection and boundary mixing can introduce significant differences in model tracer transport regardless of whether or not the models use the same reanalysis data (ECMWF or otherwise). We have been through the manuscript and made clear, where appropriate, that the treatment (or implementation) of transport processes differs between the models (though all models are attempting to simulate convection and mixing processes).

In the revised manuscript we now comment on the fact that most models use ERA-Interim (see first paragraph of Section 2.3).

*Line 136 – is aseasonal the same as 'annual average'?*

The top-down emission inventory in question was formulated using somewhat sporadic tropospheric VSLS measurements in various years and locations. The limited availability of measurements within a given year did not allow seasonality to be derived (i.e. the inventory is aseasonal). Similarly, nor can the fluxes be considered annual averages (although this *may* be a reasonable approximation).

*Section 2.1*

*It would be useful to have some visual comparison of the emissions (perhaps supplementary material)? The words don't give a sense of how large the differences in emissions are, and without that the sensitivity to emissions or lack thereof does not make sense.*

OK. Figures comparing the surface emission fields for both $CHBr_3$ and $CH_2Br_2$ is now given in a new Supplementary Information.

*Line 160 the words after the semi-colon should have a verb, or the sentence should be re-written without a semi-colon.*

OK. The sentence has been re-written accordingly.

*Line 184 'diagnosed convection' – do you mean used the standard parameterization for transport? Identified convection? Not clear.*

OK. We have made this clearer now. The "diagnosed convection" in TOMCAT is the standard scheme yes.

*Section 2.3 Did the CCMs use observed sea surface temperatures (relevant for El Nino)?*

Yes and we have now noted this in the footer of Table 2.

*Section 3.1.1 Line 300 – MHD, THD, CGO, PSA - simulated seasonal cycles do not agree with data – does the simulated seasonality look like the imposed seasonality of the loss terms? (If it does, then how can the models perform differently?)*

Yes, the simulated seasonality at these sites is consistent with that expected from the seasonality of the chemical loss. We have added a sentence in Sect. 3.1.1 making this point.

The Reviewer asks "how can the models perform differently?". On this point, first, at MHD the models in fact look very similar for both $CHBr_3$ (see Figure 3) and $CH_2Br_2$ (Figure 4). It is not surprising that at some sites there will be differences between the models as the models have different transport schemes. This is not limited to simply the large-scale resolved horizontal/vertical winds, for most models taken from reanalysis products (i.e. ECMWF), but also includes differences in parameterisations of both convection and boundary layer mixing. Differences in such will impact both absolute tracer mixing ratios at a given site and can also impact seasonality (see references below). We make this point in the revised paper in the following lines:

From revised Section 3.1.2:

"As the chemical sink of VSLS was consistent across all models, the inter-model differences discussed above are attributed primarily to differences in the treatment and implementation of transport processes. This includes convection and boundary layer mixing, both of which can significantly influence the near-surface abundance of VSLS in the real (Fuhlbrügge et al., 2013, 2015) and model (Zhang et al., 2008; Feng et al., 2011; Hoyle et al., 2011) atmospheres, and are parameterised in different ways (Table 2). On this basis, it is not surprising that different CTM setups lead to differences in the surface distribution of VSLS, nor that differences are apparent between CTMs that use the same meteorological input fields. Indeed, such effects have also been observed in previous model intercomparisons (Hoyle et al., 2011). Large-scale vertical advection, the native grid of a model and its horizontal/vertical resolution may also be contributing factors, though quantifying their relative influence was beyond the scope of TransCom-VSLS."

*Line 315 – Is it important that the observed annual variation is much smaller at SMO and CGO than at many of the other sites? Disregarding SMO (some really weird behaviour), CGO and PSA amplitude greatly overestimated although shape is vaguely similar. Any commonalities among the 60% of models that do not correlate with observations > 0.5? Resolution? Meteorology? Transport scheme? Boundary Layer dynamics? Anything?*

At mid-high latitude sites, the local lifetime of these VSLS is approximately 1 month or longer, and the effect of loss seasonality will be similar at sites within similar latitude bands; e.g. because of seasonality in OH following the solar insolation. Also note that the seasonal difference in OH concentration increase towards the poles. Since SMO is located in the tropics the seasonality in loss due to OH is weaker compared to CGO, MHD or PSA. Emission and transport can vary regionally and also have a local influence on concentration seasonality. The interactions of spatial distribution of emissions and transport are discussed later for the MHD case.

In the revised manuscript we have expanded the discussion of surface seasonality in Section 3.1.1. As previously noted we could not detangle individual transport components in each model within the framework of this large multi-model project (we note the Reviewer's opening sentence of his/her review). CGO and PSA, being outside of the tropics, are obviously far less relevant for bromine SGI than tropical locations. As SMO is within the tropics we have expanded the discussed in Section 3.1.1 on the possible causes of the model outliers.

*317 – virtually all do not reproduce – how about 'almost none of the models reproduce'
or 'virtually all of the models fail*

OK, we have changed "virtually all of the models do not reproduce" to "almost none of the models reproduce", as requested.

*I don't understand the point of this discussion (lines 317 ff) At MHD, seasonality in the local emission flux is suggested to be the dominant factor controlling the seasonal cycle of surface CHBr3 (Carpenter et al., 2005). This leads to the observed summer maximum (as shown in Figure 3) and is not represented in the models' CHBr3_L tracer which, at the surface, is driven by the aseasonal emission inventory of Liang et al. (2010). Why did Carpenter et al. make that 'suggestion'? This sort of model can only do what you tell it, so if Carpenter et al. are correct – then you would never expect the models to do this. So then, what is the point of going to the MMM?*

Of course we agree that the seasonality in the emissions will not be captured (these are aseasonal emission inventories, as noted). However, seasonality in the emission flux (and the chemical loss) may not be the only factor contributing to the observed seasonal cycle in the CHBr$_3$ surface mixing at MHD. Our results suggest that transport processes serving the MHD site are also important. MHD is served by air masses mostly of marine origin and several studies have shown a marked seasonality in the transport of air masses arriving at the site (e.g. Cape et al., 2000).

In the revised manuscript we now include individual model-measurement correlation coefficients (in addition to the MMM) at MHD (in the new Supplementary Information, Table S1). These comparisons reveal that 7 out of the 11 models capture the MHD CHBr$_3$ seasonal cycle (with r > 0.65) when using the Ziska et al. inventory, despite the inventory being aseasonal. (Note, this was also shown by Lennartz et al. 2015, ACP). We have also now discussed why this is seen for the Ziska inventory and not the other aseasonal emissions. Given that this is the only difference (i.e. the chemistry is the same between simulations in a given model), the only possible conclusion is that the distribution of emissions, with respect to transport processes, is causing this effect. We already make this point in the paper, but have expanded the discussion in Section 3.1.1 so that it is clearer (see response to point below).

Cape et al.: the use of trajectory cluster analysis to interpret trace gas measurements at Mace Head, Ireland, Atmos. Env., 2000.

*Why aren't you discussing whether an emission inventory that has a seasonal element does better?*

With the exception of the Ordonez et al. inventory at tropical latitudes only, where it has a small seasonal element (not relevant for MHD), the inventories are aseasonal. We are not aware of a prescribed emission inventory with global seasonality in the surface fluxes and therefore cannot

speculate. Besides, as noted above, 7 out of 11 models do not require seasonally-varying $CHBr_3$ emissions to capture the observed seasonality in $CHBr_3$ mixing ratios at MHD.

*It would make more sense if there was a better sense of the differences in inventories. Specifically – why would TWO aseasonal inventories give different answers at MHD, if the seasonality of the emissions is speculated to be a controlling factor?*

We have added to the new Supplementary Information figures showing surface $CHBr_3$ and $CH_2Br_2$ emissions for each inventory (Figure S1 and S2) to provide a better sense of the differences in the inventories, as requested. It can be clearly seen that surface $CHBr_3$ emissions are much larger in the region of MHD, and indeed in the NH in general, for the Ziska et al. inventory. This has been previously highlighted in our recent work (Hossaini et al. 2013). In addition, we have added a supplementary figure (S3) comparing the absolute agreement between each model and measured surface $CHBr_3$ at MHD for Liang et al. emissions and Ziska et al. emissions. The figure clearly shows that the larger emissions in the latter provide much better model-measurement absolute agreement (for the majority of models). Regarding the seasonal cycle, clearly a comprehensive trajectory analysis of air masses arriving at MHD is well beyond the scope of this work, and is not required to support our main conclusions concerning bromine SGI (mainly occurring in the tropics). However, we have extended the discussion in Section 3.1.1 so that the above points are clearer. We now suggest that the summertime transport of air that has experienced relatively large $CHBr_3$ emissions north to north-west of MHD is the cause of the seasonal cycle seen in the Ziska et al. simulations. To further support this, animations of the seasonal evolution of surface $CHBr_3$ have been created and have also been added to the paper as Supplementary Information.

*Section 3.1.2 338 between a model value (M) and an observation (O), why parenthesis around 'for each model tracer'?*

OK. We have removed the parentheses.

*Figure 6 – ok these are the minimum percentages – but how does the reader know that the difference between a 'best' comparison and a comparison with one of the other inventories is significant?*

That is not the point of Figure 6. The point is to show which emission inventory performs best and to give the MAPE for that inventory. While it is true from Figure 6 that one cannot discern how the 2$^{nd}$ best inventory performs (in terms of MAPE) this is less relevant. Inclusion of that level of detail in *this* Figure would add unnecessary convolution to an already detailed figure and detract from the main message regarding the best inventory. Besides, one can see how the inventories perform against each other, within a given model, in the tropics (most important region for VSLS injection into the stratosphere) from Figure 7.

*350 I presume you don't get MAPE for both species with the same inventory because loss processes are different time scale? Replace: 'low CHBr3 MAPE (good agreement), at a given location using a particular inventory does not necessarily mean a corresponding low CH2Br2 MAPE can be achieved using the same inventory, at that location'. At a given location low CHBr3 MAPE (good agreement) does not necessarily accompany a corresponding low CH2Br2 MAPE using the same inventory*

Yes. We have reworded the sentence as the Reviewer suggests.

*355 – is this also related to how the inventories are created in the first place – i.e., how much do the inventories themselves depend on models and/or ERA-Interim?*

A brief discussion of the inventories is already given in Section 2.1. We also refer the reader to our recent paper that further describes the inventories in the following lines:

"As these inventories were recently described and compared by Hossaini et al. (2013), only a brief description of each is given below".

The top-down inventories are derived using aircraft observations in conjunction with models. Therefore, to some degree the derived fluxes will depend on the details of the transport scheme in the model used. However, we cannot say anything quantitative regarding this and this type of analysis is well beyond the scope of this paper. It is also not required to support our conclusions. Neither of the two top-down inventories were derived in models that use ERA-Interim.

*365 – you attribute all differences to physical processes – e.g., convection and boundary layer mixing. Since most of these use assimilated meteorology, does that mean implementations differ among CTMs. Also, there are some pretty large differences between a free running simulation and a nudged simulation, so differences among 'variants' should not be surprising.*

Yes, the differences are attributed to both the convection and boundary layer schemes in these models. Implementation of ERA-Interim is not necessarily the key or only point here, rather the fact that these models use different parameterisations of the above sub-grid processes in important. In other words, simply because a number of the models read ERA-Interim fields (horizontal/vertical winds, temperature and humidity), tracer transport will be different because of the way the above sub-grid scale processes are treated. We already include several citations to modelling work which has shown large differences in the simulated near-surface abundance of short-lived tracers, depending on the choice of convection and boundary layer transport schemes.

In addition, as noted earlier, we have now (i) defined what we mean by "transport" in the Introduction, (ii) amended the text in numerous places to clarify that implementation of transport processes is important and (iii) added the boundary layer and convection scheme used by each model to Table 2.

Regarding the variants, we removed the word "even" from the following sentence so that is sounds less surprising.

"At some sites, differences among emission inventory performance are  apparent between model variants that, besides transport, are otherwise identical."

*Finally, in the prior modelling studies that had best agreement with different inventories, the loss terms were presumably different.*

Yes, that is correct. We now make that point in Section 3.12.

*370 – why are differences in model variants surprising? In one case, this is the difference between free-running and nudged, and it is more likely than not that convection differs between these two in both intensity and location. In the second case, the chief difference that is discussed in convection, so again, performance is more likely to be different than it is to be the same.*

As already answered above, regarding the variants, we removed the word "even" so that the sentence sounds less like the result is surprising.

*395 It would be better to say 'For the N (fill in number) models that submitted hourly output . .*

OK. We have done this.

*After that, paper says "Generally, the models reproduce the observed mixing ratios from SHIVA well, with a MMM campaign MAPE of 25% or less for both VSLS." This good agreement clearly depends on who is looking, and whether it makes sense to compute MAPE for the multimodel mean when the spread indicated by shading can be as much as 1 ppt (lowest value) and about 2 ppt (MMM) (top panel).*

It is not clear what the Reviewer is asking for. It is common practice in large multi-model assessments to include multi-model mean (MMM) values and to indicate the model spread. We have done both of these things in Figure 8 (and 9). For $CHBr_3$, we believe a MAPE of <25% on the multi-model mean is good. Inclusion of more (unnecessary) data on these figures would distract from the main point that the models perform reasonably well at the surface in the tropical West Pacific, giving confidence in the corresponding modelled bromine SGI.

*It is also confusing since each model is using its 'preferred' inventory, and seriously in the real world there is only one actually set of emissions. In the best of circumstances, I think the MMM conceals physical differences and/or deficiencies in a subgroup of models. Here, with each model using its 'preferred' inventory, I think it is nearly impossible to understand they significance of good or poor agreement with the MMM.*

As noted, it is common practice in large multi-model assessments to include MMM values. Naturally it will be the case that some models look better than others and that the MMM does not provide that information. However, we already provide the model spread (as is also common practice) and most comparisons throughout the paper show results from individual model profiles. The point on "preferred" inventory is dealt with below.

*Section 3.2 412 – by using the model 'preferred' inventory, what you are testing here is given surface values, how similar is the transport to higher levels to that inferred from observations in the real atmosphere. There are other ways to do this of course – in fact, looking at EACH tracer profile as a fraction of its near surface value might be even more instructive. Nonetheless – the discussion is convoluted and should be re-written to state the main (physical) point clearly. I presume 'paramerized transport schemes' later in this paragraph refers mainly to convection?*

Yes. A major goal of this work was to provide the first multi-model estimate of the climatological SGI of bromine from VSLS reaching the stratosphere. To accomplish this, the preferred tracer approach was needed. It would not be correct to make predictions of how much $CHBr_3$/$CH_2Br_2$ enters the stratosphere, from a given model, using an emission inventory that does not provide that model with good or reasonable agreement to measured surface $CHBr_3$/$CH_2Br_2$. Use of the optimal/preferred inventory in this regard is essential. We note, a major and novel finding of this work is that the models do not necessarily agree as to which inventory performs the best in many locations.

We have reworded the sentence beginning "This approach ensures.." in Section 3.2 for clarity. Directly after this we have now stated what we are testing, as suggested by the Reviewer.

Yes, we are referring to sub-grid scale transport schemes, such as convection. We have now added "sub-grid scale" for clarity.

*447 Only the number of flights controls the variability comparing Pre-AVE to CR-AVE? Nothing seasonal or spatial? Are the models sampled like the aircraft to produce average profiles? Is the error bar the range of values, the standard deviation? Would standard error of the mean be better? The correlation coefficient – is that the correlation for the whole profile? Isn't that guaranteed to be large since observed and simulated profiles general decrease with altitude?*

Pre-AVE and CR-AVE were broadly in the same region and in the same months (already clear from Figure 2 and from the campaign descriptions in Section 2.4.2). We therefore anticipate that the sample size is a significant contributor to the variability.

Yes. The models are sampled like the aircraft and then averaged.

The error bars on Figure 11 are the standard deviation as indicated in the caption. For consistency we have used standard deviation throughout the paper.

Yes, the correlation coefficient is over the whole profile. It is not guaranteed to be large since variability can be significant (see e.g. panel c of Figure 11). It is a good indicator of model skill that the models reproduce the decrease with altitude and that correlation coefficients are large.

*ATTREX higher values at higher altitude 'possibly reflects the location'? Isn't this true (and backed by other observations?) If it is only 'possible', what are the other causes?*

Yes, OK, we have removed "possibly".

*grammar - CR-AVE had nearly twice the number of flights AS Pre-Ave and . . .*

OK, we have changed "than" to "as".

*Section 3.3*
*470 ' likely reflects the location at which the measurements were made' Why so many words, why 'likely' (what else could it be) and why no direct statement about zonal asymmetry? Would the model zonal means compare better with Carpenter and Reimann? Or should it be model mean in a different region compared with Carpenter and Reimann?*

OK. We have removed "likely". We do not follow the 2$^{nd}$ part of the Reviewer's comment. The last paragraph of Section 3.3 discusses measurements of $CHBr_3$ and $CH_2Br_2$ from ATTREX (not models) versus those from the WMO compilation.

*Section 3.4*
*515 If most of the models are using assimilated fields, how can they fail to locate the areas of deep convection and the seasonal dependence therein? It is all right to describe this behaviour, but I would hardly call it a prediction. Too much discussion, especially since the result is not novel.*

Perhaps the Reviewer is asking a rhetorical question here. As noted earlier, the treatment of convection in models can vary significantly. Use of meteorological fields from reanalysis data (e.g. ECMWF) does not guarantee a "good" simulation of deep convection, though clearly accurate fields of temperature, pressure, humidity and wind fields (from the reanalysis), which feed into convective parameterisation, are desirable. Such parameterisations are often complex (with many tuneable parameters) and make assumptions, for example, on whether shallow, mid-level or deep convection takes place in a column and the trigger threshold for such. Clearly, the point of this paper is not to provide a comprehensive critique of the parameterisation of convection in 11 models. We already make the point in the introduction that "While global models generally simulate broadly similar features in the spatial distribution of convection, large inter-model differences in the amount of tracers transported to the tropopause have been reported…". It is unclear where the Reviewer is referring when he/she says there is too much discussion. On line 515, where he/she states, there is only a very brief description that the models show the largest $CHBr_3$ mixing ratios at the tropopause over the West Pacific in DJF.

*525 – variations in the importance of Monsoon – any connection to the input data or model type? I don't think this is evidence that UKCA-HI has a more faithful representation of convection – you would need some other information about HI vs LOW to make this statement.*

No clear relationship between the importance of the Monsoon and the input data/model type could be discerned from this analysis (this is not required to support our main conclusions). This would require a more detailed and dedicated examination of the Monsoon region, outside the scope of the TransCom-VSLS framework. We now make this point in Section 3.4 of the revised manuscript.

Regarding UKCA-HI vs LOW, we agree that the statement was too strong and have reworded.

*Section 4*

*605 previously when you talked about variability it was physical (e.g., seasonal etc. –something real). Here you are talking about differences among models for different inventories. It is confusing to call*
*this 'variability'.*

We agree and have changed "inter-model variability to "inter-model differences".

*622 – model variants are identical except for tropospheric transport schemes. Based on everything else written, I don't think this statement is correct. E.g, the tropospheric transport of 'nudged' vs 'free-running' will differ for physical reasons, not just 'transport schemes'.*

Yes, we agree that is the case for EMAC_N vs free-running EMAC_F. However, for TOMCAT and TOMCAT_CONV (offline CTMs), the models are identical apart from the convective transport schemes. Therefore, we have replaced:

"This effect was even observed between model variants which, other than tropospheric transport schemes, are identical".

With:

"This effect was also observed between CTM variants which, other than tropospheric transport schemes, are identical".

*The problem with single model studies of inventories or deriving inventories is that they don't typically include model error. If they did then the inventories would be more robust – or the differences among studies would likely fall within the errors.*

We agree.

*625 – For both CHBr3 and CH2Br2 the 'best' inventory for the tropics is the lowest – but at the same time agreement here is 'less sensitive to choice of inventory'. What point are you trying to make? I don't see how the statements about seasonally resolved air-to-sea fluxes follow from anything in this paragraph (noting this is the 'discussion and conclusions' section).*

OK, this was not clear. The "less sensitive to choice of inventory" refers to $CH_2Br_2$ not the tropics. We have reworded this sentence to clarify.

The inventories we currently use are aseasonal. We feel it is good to make the point that a move towards seasonally-resolved VSLS emissions in models would be a good direction to go in. We feel as though this fits in following the discussion of surface emission inventories.

*665 – the very long sentence beginning 'Although . . . ' should be clarified.*

OK, we have shortened this sentence and removed unnecessary words to improve clarity.

*Picky comment Example: Do you really need to include so many references – e.g., five references to say that Bromine + chlorine destroys ozone more than chlorine by itself?*

OK, we removed one of the references on this point from the introductory paragraph.

---

## Author Comment (AC2) · 3 May 2016

Our responses to review comments (repeated in *italics*) are given below in **red**.

**Response to Reviewer 2**

*This paper presents a comprehensive model intercomparison of the impact of bromine containing VSLS on the stratospheric bromine loading. This is a good initiative and the outcome of this intercomparison will be important of assessing the impact of bromine on stratospheric ozone. It is in particular noteworthy that a lot of observations are employed to assess the quality of the model results.*

We thank the Reviewer for his/her comments on our manuscript. We are pleased that he/she finds the work important and acknowledges the large body of atmospheric observations used throughout.

*I have some points (see below), where I think the discussion in the paper can be clarified and improved. The impact of particular model features (e.g., the convective schemes employed) on the different results could be brought out more clearly. The reader ultimately will be interested in what the problematic model features are, because these are the features that need improvement in the further developments of such models. This point cold be brought across in the paper in a better way. In summary, I think that a revised version of the paper, taking into account the points raised in the reviews will be a valuable contribution to ACP.*

We have addressed the review comments and believe the paper has been strengthened accordingly. We are pleased the Reviewer thinks this work will make a valuable contribution to ACP.

**Detailed comments**
*Five out of the 11 participating models are nudged to or driven by ERA-Interim. While ERA-Interim is a good choice, this fact will lead to the multi-model mean being biased to an ERA-Interim world. I suggest to bring this point across more clearly. Does this fact have any implications for the conclusions of this model intercomparison?*

It is indeed the case that most models use ERA-Interim met fields, but as you can see other models (ACTM, NIES or the free running models) do not produce MAPE very different from those produced by the ERA-interim driven model. In fact there is quite good agreement between the major reanalysis products from ECMWF, JMA and NCEP. This was very recently highlighted by Harada et al., (2016) and is clearly shown in Figure 4 of that paper: see: http://jmsj.metsoc.jp/EOR/2016-015.pdf. Thus we do not believe at this stage that our model spread is hugely biased towards ERA-interim. In the revised manuscript we have commented on this, as requested, in the first paragraph of Section 2.3.

[Figure]

Figure 4. Annual and seasonal mean residual-mean upwelling within 20° of the equator, averaged

from 1979 to 2012.

*Figure from Harada et al., JMSJ, 2016*

*Another model feature, which is important for tropospheric transport of VSLS is the convective parametrisation used in the model (see for example Rybka, H. and Tost, H.: Uncertainties in future climate predictions due to convection parameterisations, Atmos. Chem. Phys., 14, 5561-5576, doi:10.5194/acp-14-5561-2014, 2014, and references therein). I suggest more discussion of this point in the paper. Also, the information of the convective scheme used in the different models should be included in Table 2. Perhaps some of the model differences and some of the model similarities can be attributed to using a particular convective parametrisation or a particular meteorology?*

As per the Reviewer's suggestion, we have extended Table 2 to include the convection and boundary layer mixing schemes used by each model. We did not find any systematic differences in our results related to the choice of convection scheme or input meteorology. We have now made this point in the abstract:

"Overall, our results do not show systematic differences between models specific to the choice of reanalysis meteorology, rather clear differences are seen related to differences in the implementation of transport processes in models".

Related to the above, and following comments from Reviewer #1, we have made clear that differences between CTM setup and the implementation of transport processes is important, as all models would claim to be simulating the above physical processes. Further, we now define what we mean when referring to "transport" differences early on in the manuscript (last paragraph of Introduction):

"... we define *transport* differences between models as the effects of boundary layer mixing, convection and advection, and the implementation of these processes. Note, the project was not designed to separate clearly the contributions of each transport component in the large model ensemble, but can be inferred as the boundary layer mixing affects tracer concentrations mainly near the surface, convection controls tracer transport to the upper troposphere and advection mainly distributes tracers horizontally (e.g. Patra et al. 2009)"

Finally, we have expanded the discussion of the role of convection in the manuscript. In addition to new text placed in Section 3.1.2 (suggested by Reviewer #1), we have added discussion in Section 3.4 relating to model differences shown in Figures 14 and 15. We also cite the *Rybka and Tost* paper.

"The high altitude model-model differences in $CHBr_3$, highlighted in Figures 14 and 15, are attributed predominately to differences in the treatment of convection. Previous studies have shown that (i) convective updraft mass fluxes, including the vertical extent of deep convection (relevant for bromine SGI from VSLS), vary significantly depending on the implementation of convection in a given model (e.g. Feng et al., 2011) and (ii) that significantly different short-lived tracer distributions are predicted from different models using different convective parameterisations (e.g. Hoyle et al., 2011). Such parameterisations are often complex, relying on assumptions regarding detrainment levels, trigger thresholds for shallow, mid-level and/or deep convection, and vary in their approach to computing updraft (and downdraft) mass fluxes. Furthermore, the vertical transport of model tracers is also sensitive to interactions of the convective parameterisation with the boundary layer mixing scheme (also parameterised) (Rybka and Tost, 2014). On the above basis and considering that the TransCom-VSLS models implement these processes in different ways (Table 2), it was not possible to detangle transport effects within the scope of this project. However, no systematic similarities/differences between models according to input meteorology were apparent".

*I also have reservations about the concept of a "preferred" tracer. I think this means that the emission inventory somehow interacts with the transport scheme of the model to produce reasonable results at higher altitudes. But this means that the higher altitude agreement could be right for the wrong reason. I know it is demanding a lot from models, but of course one would*

*expect to design independently the best emission inventory and the best (vertical) transport to obtain the best agreement with measurements. Obviously this model intercomparison cannot achieve this goal, but I think the discussion of these issues could be improved.*

The overarching goal of the work was to calculate a climatological multi-model best estimate of stratospheric bromine SGI from $CHBr_3$ and $CH_2Br_2$. It was essential, of course, for this estimate to be based on simulations that provide the best possible model-measurement agreement at the surface. Given good surface agreement, the models' transport of $CHBr_3$/$CH_2Br_2$ from the surface to higher altitudes, against that observed, has been tested. The fact that models do not necessarily agree as to which emission inventory "performs best" at all surface sites (against measurements) we believe is an important finding of this work. It has implications for model studies attempting to quantify the global flux of VSLS to the atmosphere and, in particular, for studies attempting to reconcile such estimates obtained from different models. We have added a sentence to the revised manuscript in Section 3.1.2 (end of 3$^{rd}$ paragraph), where the above is discussed, to make the latter point more clear:

"Ultimately, attempts to reconcile estimates of global VSLS emissions, obtained from different modelling studies, need to consider the influence of inter-model differences, as discussed above."

The discussion of "preferred tracer" has also been clarified in Section 3.2 in response to a comment from Reviewer #1.

*Finally, the impact of ENSO activity on the stratospheric bromine loading is unclear. What is the message of the paper here? The paper states that there is a strong correlation of SGI with ENSO (e.g. abstract), but that there is no correlation of ENSO (MEI) with the bromine loading in the LS (e.g. conclusions). But SGI is important for the bromine loading in the LS. This points needs to be clarified and better discussed in the paper.*

OK, we have clarified this. Our results show that (i) SGI is enhanced over the East Pacific during strong El Niño conditions (e.g. in 1997/1998 as can be seen in Figure 17). Related to this (ii) SGI is strongly correlated to MEI over the East Pacific (where significant SST warming occurs under El Niño conditions) but (iii) averaged over the whole tropical domain, there is little correlation between SGI and ENSO. The latter point is because of the zonal structure in SST anomalies (and therefore convective activity) associated with ENSO activity. Essentially, the effect of warming and intensified convection in some areas (i.e. East Pacific) on stratospheric Br SGI can be cancelled out by the cooler SSTs in other tropical regions. Aschmann et al. (2011) performed a detailed analysis of how bromine SGI is affected by ENSO and indeed reported this complex zonal structure. We have now clarified these points at the end of Section 3.5 of the revised manuscript.

**Minor issues**
*• Title: I am not sure if "TransCom-VSLS" should be in the title; the name of the project will not be relevant on a timescale of years, when the paper will still be read.*

Since the concept and experimental method are chosen from previous TransCom experiments, it gives a link to the paper's evolution. Thus we would prefer to keep this term in the title.

*• l. 7: I do not think that model estimates should be used to "constrain" measurements.*

We are referring here to using models to help constrain the current SGI range.

*• line 20: change 'optimal' to 'best'*

OK. We have done this.

*• l. 36: Isn't 6 month a bit long for very short lived?*

We agree that intuitively it does seem long for a "very short-lived" compound. However, this is the definition used in previous WMO Ozone reports. VSLS local lifetimes at the surface can vary substantially in space and time, though the <6 months rule is broadly accurate.

*• l 51: 'recent' twice in this sentence*

OK. A "recent" has been removed.

*• l. 52: try nmathrm{VSLS} to avoid italics in VSLS. (Similar for MAPE (l. 345) below).*

We thank the Reviewer for this suggestion. The italics have been removed from "VSLS" in $Br_y^{VSLS}$ throughout the manuscript.

*• l. 59: 'owing to' instead of 'due to'*

OK, we have replaced this.

*• l. 76: I think you mean Tissier and Legras here*

Yes, that has been corrected.

*• l. 78: do you mean "broadly similar" here?*

Yes, we have corrected this.

*• l. 100: what do you mean by "climate modes" – more explanation here.*

We are referring to modes of climate variability, specifically ENSO in this case. We have been more explicit in the revised manuscript and refer directly to ENSO rather than "climate modes".

*• Figure 1: This figure is not really discussed in the paper. Which message does it communicate? I suggest removing the figure from the paper.*

We feel that Figure 1 provides a visual overview of the experimental design for the reader. Otherwise it is extremely difficult to show the flow of work and the design of experiments, with several emission scenarios, to a new reader. In this figure we also show how the model output is used for calculating SGI. Some of these concepts are new to the TransCom initiative, thus we would prefer to keep it.

*• l. 144: is a bottom-up . . .*

We have changed "bottom-up" to "a bottom-up".

*• line 179: this means that the multi-model mean is highly influenced by CTMs driven by ERA-Interim data – correct?*

Yes. We have now commented on this in the manuscript. Please see our above response to the first detailed review comment on page 1 of this document.

*• l. 211: instead of 'see also' you could perhaps state for which information which paper should be consulted.*

We have removed the reference following "see also" as this was not needed.

*• l 301: what is the reason that 'clear outliers' are found? Are these models with obvious errors?*

The text here is discussing Figure 3. Generally there are few outliers and these outliers are limited to specific sites (for example, B3DCTM and STAG at SMO). We have added some text directly following the sentence containing "clear outliers" in Section 3.1.1 commenting on potential causes.

"The cause of the outliers at a given site are likely in part related to the model sampling error, including distance of a model grid from the measurement site and resolution (as was shown for $CO_2$ in Patra et al., 2008). These instances are rare for VSLS but can be seen in B3DCTM's output in Figure 3 for $CHBr_3$ at SMO. B3DCTM ran at a relatively coarse horizontal resolution (3.75°) and with less (40) vertical layers compared to most other models. Note, it also has the simplest implementation of boundary layer mixing (Table 2). This above behaviour is also seen at SMO but to a lesser extent for $CH_2Br_2$, for which the seasonal cycle is smaller (see below). The STAG model also produces distinctly different features in the seasonal cycle of both species at some sites (prominently at CGO, SMO and HFM). We attribute these deviations to STAG's parameterisation of boundary layer mixing, noting that differences for $CHBr_3$ are greater at KUM than at MLO – two sites in very close proximity but with the latter elevated at ~3000 metres above sea level (i.e. above the boundary layer)."

Later in the paragraph: "The NIES-TM model does not show major differences from other models for $CHBr_3$, but outliers for $CH_2Cl_2$ at Southern Hemispheric sites (SMO to SPO) are apparent. We are unable to assign any specific reason for the inter-species differences seen for this model".

• l. 329: use r for the correlation coefficient

OK, we have italicized "r" throughout the text.

• l. 366: why does convection influence "near-surface" abundances of VSLS?

Convection lofts tracer mass away from the boundary layer. We already point to several references which show this to be the case. For example, see plot of convective updraft mass fluxes in Figure 3 of Feng et al. (2011) which shows the vertical extent of convection.

• l. 414: I think it is problematic that models have a preferred tracer: doesn't this imply that results could be right for the wrong reason?

This comment is addressed in the detailed comments section above.

• l. 425: Where is the reproduction of the c-shape shown? This seems an important issue.

It is clearly visible for SHIVA and HIPPO-1. We have now been explicit as to where we are referring: panel (a), 2nd and 3rd row of Figure 10.

• l. 435: The concept of a 'preferred' tracer means that the emission inventory somehow interacts with the models transport scheme to produce reasonable results at higher altitudes – correct? Can you describe in more detail here, what 'worse agreement' means?

The models, with good agreement at the surface (by way of their preferred tracer), produce a sound simulation of the transport of $CHBr_3/CH_2Br_2$ from the surface to higher altitudes (as evidenced in Figures 10 and 11). The point we are making is that if the model-measurement agreement at the surface is degraded (e.g. in the simulations using the non-preferred tracer), then the absolute model-measurement agreement at higher altitudes is also worsened. We have reworded the text here for clarity. The sentence now reads:

"For a given model, simulations using the non-preferred tracers (i.e. with different $CHBr_3/CH_2Br_2$ emission inventories, not shown), generally lead to worse model-measurement agreement in the TTL. This is not surprising as model-measurement agreement at the surface is poorer in those simulations."

• l. 485: is CO really short-lived?

CO has a global lifetime of several months and therefore is similar to some VSLS. In order that CO and VSLS are not confused, we have reworked this sentence.

• l. 492: state the lifetime in months/weeks

OK, we now state that $CH_2Cl_2$ has a local lifetime > 1 year in the TTL.

• l. 527: you might want to add here also Tissier and Legras 2015; Vogel et al. 2014

OK. We have done this.

• l 560: Clarify which best estimate is meant here, TransCom or WMO.

WMO. We have added the citation to Carpenter and Reimann to clarify.

• l. 593-595: The last sentence states that the VSLS loading in the LS is not correlated to MEI. But the sentences above state that bromine SGI is sensitive to modes such as MEI. Isn't this a contradiction? I think more discussion is require here.

See earlier and also later comment. We have clarified this section of text and we are saying that the correlation is related to a particular region (the tropical E Pacific).

• l. 598: change to: these processes

OK, we have added "processes".

• l. 599: change 'a range' to 'a number'

OK, we have done this.

• l. 614-618: Is the point here that the seasonal cycle is not dependent on the emission inventory, but the absolute model-measurement agreement is? How can this be the case. Please clarify. (See also abstract).

Yes, that is correct and is the case because at most sites the seasonal $CHBr_3/CH_2Br_2$ abundance is determined from seasonality in the chemical loss rate (same for all models and same between model simulations with different emissions). We have clarified this point. The sentence now reads:

"At most sites, (i) the simulated seasonal cycle of these VSLS is not particularly sensitive to the choice of emission inventory, and (ii) the observed cycle is reproduced well simply from seasonality in the chemical loss (a notable exception is at Mace Head, Ireland)."

Of course the absolute model-measurement agreement will be sensitive to the emission fluxes as they vary in strength.

• l. 626: change optimal to best

OK, we have done this.

• l. 634: what exactly is meant by 'online calculations'?

An "online" emission calculation here refers to one in which emissions are calculated by taking into account the interaction between the atmospheric state and the ocean. Online calculations consider the actual seawater concentration of VSLS to derive air-sea concentration gradients and calculate

fluxes. This is different to the approach mostly used to date whereby climatological emissions are prescribed. We have clarified this in the text.

• l. 648: But the 'higher altitudes' are most relevant for the transport of VSLS into the stratosphere – correct?

Yes. The model-measurement agreement during ATTREX is discussed in Section 3.3. We have added a sentence in the Summary and Conclusions noting that most models fall within 1 standard deviation of the observed mean at the tropopause.

• l. 663: You mean the SGI range by Carpenter and Reiman, add the citation for clarification.

OK, we have done this.

• l. 670-672: This is astonishing, isn't it? I suggest somewhat more discussion on this point.

This comment was answered earlier. Essentially, it may not be too surprising as changes to SSTs (and convective activity) associated with ENSO is zonally very asymmetric, with warming in some regions and cooling in others. The warming of East Pacific SSTs under El Niño conditions leads to enhanced SGI over this region, but when SGI is averaged over the whole of the tropics, this effect is dampened/cancelled. We have clarified these points at the end of Section 3.5

• l. 676: change 'changes to' to 'changes of'

We feel that "changes to emissions" reads better than "changes of emissions".

• l. 678: change 'increased' to 'increase of the'

OK, we have done this.

• l. 679: distinguished from what?

From the present day loading. To clarify we have changed "distinguished" to "determined".

• l. 689: why is R Hommel not abbreviated?

We did not abbreviate R Hommel because the abbreviation would be the same as the earlier, but different "RH".

• Fig. 1: not sure if this figure is necessary

We feel that Figure 1 provides a visual overview of the experimental design for the reader. We would prefer to keep it. Perhaps the Editor can comment on this.

• Fig. 2: Continents in light grey would look better than in black.

OK, we will update.

• References: There are some references that need to be updated; ACP vs ACPD, Werner et al., 2016 etc.

We have updated the reference list.